# A conserved RNA structural motif for organizing topology within picornaviral internal ribosome entry sites

Deepak Koirala [1], Yaming Shao[1], Yelena Koldobskaya[1], James R. Fuller [1], Andrew M. Watkins[2], Sandip A. Shelke [1], Evgeny V. Pilipenko[1], Rhiju Das [2], Phoebe A. Rice[1] & Joseph A. Piccirilli [1,3]

Picornaviral IRES elements are essential for initiating the cap-independent viral translation. However, three-dimensional structures of these elements remain elusive. Here, we report a 2.84-Å resolution crystal structure of hepatitis A virus IRES domain V (dV) in complex with a synthetic antibody fragment—a crystallization chaperone. The RNA adopts a three-way junction structure, topologically organized by an adenine-rich stem-loop motif. Despite no obvious sequence homology, the dV architecture shows a striking similarity to a circularly permuted form of encephalomyocarditis virus J-K domain, suggesting a conserved strategy for organizing the domain architecture. Recurrence of the motif led us to use homology modeling tools to compute a 3-dimensional structure of the corresponding domain of foot-and-mouth disease virus, revealing an analogous domain organizing motif. The topological conservation observed among these IRESs and other viral domains implicates a structured three-way junction as an architectural scaffold to pre-organize helical domains for recruiting the translation initiation machinery.

[1] Department of Biochemistry and Molecular Biology, The University of Chicago, Chicago, IL 60637, USA. [2] Department of Biochemistry, Stanford University School of Medicine, Stanford, CA 94305, USA. [3] Department of Chemistry, The University of Chicago, Chicago, IL 60637, USA. Correspondence and requests for materials should be addressed to J.A.P. (email: jpicciri@uchicago.edu)

RNA structures in the 5′ and 3′ untranslated regions (UTRs) of an mRNA play fundamental roles in initiating and regulating translation, and influence nuclear export, cellular localization and the stability of the transcript[1–3]. For most eukaryotic mRNAs, translation initiation involves the interaction of translation initiation factors with a 5′-cap structure in the mRNA followed by the recruitment of the 40S ribosomal subunit, which then scans the mRNA to find a suitable start codon. In contrast, many viral genomes and a subset of cellular mRNAs bypass this canonical mechanism of translation initiation and use a non-canonical, cap-independent mechanism that involves the cis-acting RNA elements. These RNA elements located at the 5′-UTRs are usually known as internal ribosome entry sites (IRESs)[2–5]. During translation initiation, RNA domains in an IRES either interact with the 40S ribosome directly or recruit the ribosome through the interaction with the translation initiation factors in a cap-independent manner[2–8]. Similarly, cap-independent translation of many plant viruses involves RNA elements located near or within the 3′-UTRs of their genomes, termed cap-independent translational elements (3′-CITEs)[9–11]. Despite significant differences in size and location within viral genomes compared to IRES elements, 3′-CITEs essentially play roles analogous to IRES elements in recruiting translation initiation factors or the ribosome subunits[9–11]. The CITEs "circularize" the viral genome presumably by base-pairing interactions with the 5′ end, thereby priming the genome for translation initiation[9–11]. Here, we report the structure of a key RNA domain from an IRES that likely interacts with initiation factors, domain V of the IRES of hepatitis A virus (HAV), which is a member of *hepatovirus* genus from *picornaviridae* family.

Despite a conserved biological function, IRESs diverge significantly in their primary sequences, secondary structures, and requirements for trans-acting translation initiation factors. For example, in hepatitis C virus (HCV) or cricket paralysis virus (CrPV) the IRES recruits the ribosome directly without the pre-recruitment of translation initiation factors, whereas many picornaviral IRESs recruit the ribosome through their interaction with translation initiation factors[6–8,12–15]. Such diversity occurs even within the *picornaviridae* family, and thus picornaviral IRESs have been classified further into five types based on their predicted RNA secondary structures and their translation factor requirements, with the HAV IRES studied here belonging to type III. Moreover, some picornaviral IRESs such as those classified into type IV have secondary structure elements more similar to the IRES of HCV from *Flaviviridae* family than to IRESs within their own family. Consequently, correlating the structural organization of RNA domains in various types of picornaviral IRESs to their biological role remains a challenge, limiting our understanding of the mechanism of IRES dependent translation compared to the canonical translation.

Many biochemical, biophysical, and structural probing approaches have been used to elucidate the secondary structure of IRES domains and to understand the interaction of those domains with their cognate initiation factors[16–20]. Recent high-resolution structures of dicistrovirus and HCV-like IRESs in complex with the ribosome, enabled by advances in cryo-electron microscopy, have yielded significant insight into how IRES structure relates to its biological function[6–8]. The ability of these IRES elements to recruit the ribosome directly has facilitated the acquisition of particles appropriate for structural studies with cryo-electron microscopy. In contrast, little high-resolution structural data has emerged for picornaviral IRESs which contain relatively long, highly flexible RNA elements[21] and engage the translation initiation factors and the ribosome via a multistep, dynamic assembly process[6–8,12–14]. For picornaviral type II IRESs, a recent NMR structure of the J–K domain from the encephalomyocarditis virus (EMCV) represents the only high-resolution structure reported to date[22]. Here, we investigated the crystal structure of the corresponding domain of a type III picornaviral IRES–dV of HAV[23–25]. Compared to picornaviral type I and type II IRESs, the HAV IRES drives translation relatively inefficiently and imposes distinct requirements on cellular translation initiation factors[26,27], which has rendered study of HAV IRES more challenging.

The IRES studied here occurs within the 7.5 kb positive-sense ssRNA HAV genome, which consists of a single open reading frame flanked by highly conserved 5′ and 3′ UTRs[28]. The ~735-nt 5′-UTR contains six modular domains comprising highly organized RNA secondary structures that include the IRES elements designated as domains II to VI (Fig. 1a and Supplementary Figs. 1 and 2)[29]. Biochemical studies have shown that although domains II and III enhance translation, domains IV and V constitute the core of the HAV IRES sufficient to initiate translation by themselves[23]. Domain V resides upstream of the pyrimidine-rich tract before the AUG initiation codon (Fig. 1a). Based on its location within the IRES and the role of biochemically well-studied, analogously positioned domains from type I and type II picornaviral IRESs, domain V likely contributes to recognition of initiation factors for recruitment of the ribosome during translation initiation[14,16,17,23,30]. For example, several studies have shown that corresponding domains from type I (PV domain V) and type II (EMCV J–K domain) IRESs both interact with the initiation factor eIF4G. Moreover, a recent biochemical and biophysical study suggested that polio virus (PV) and HAV IRES domains may share a common mechanism of binding and utilizing translation initiation factors such as eIF4G and eIF4F for viral translation[31]. However, it is not clear how these domains might perform similar roles during the viral translation despite having highly dissimilar primary sequences and predicted secondary structures.

We employed a chaperone-assisted RNA crystallography approach to obtain the high-resolution structure of dV from HAV IRES. We obtained an antibody fragment (Fab) that binds to the RNA with low nanomolar affinity, crystallized the RNA in complex with the Fab and solved its structure at 2.84-Å resolution using the Fab as a molecular replacement model for the initial phasing. Unexpectedly, we observed that this IRES domain folds into a Y-shaped structure that topologically resembles the previously reported NMR structure of the J–K domain from EMCV IRES[22]. In particular, these domains exhibit striking similarities in the organization of the three-way junction by an analogous motif with an A-rich loop closed by a lone pair stem[32] (designated A-rich motif or $A_L$ hereafter) and in the location of the bulges within the analogous helices. Such structural homology allowed us to obtain a three-dimensional structural model for the corresponding domain (dIV) in foot-and-mouth disease virus (FMDV) IRES using computational approaches. Consistent with the previous biochemical observations[14,16,17,33], the FMDV dIV structure revealed a topological homology with EMCV J–K domain or HAV dV, including the similar structural configuration within the three-way junction. The helical junction architecture organized by an $A_L$ motif within these three-way junctions approximately resembles those suggested by computational models for some 3′-CITES but organized by a pseudoknot[10,34]. Given the diversity of primary sequences and secondary structures among picornaviral IRESs and 3′-CITEs, such similarity has shed light on the presence of analogous architectural strategies for organization of RNA domains that recruit translation initiation factors. Moreover, the dV structure, which represents the first high-resolution crystal structure of an RNA domain from any picornaviral IRES, will facilitate efforts to understand the structural organization and functional roles of essential RNA domains within the picornaviral IRESs.

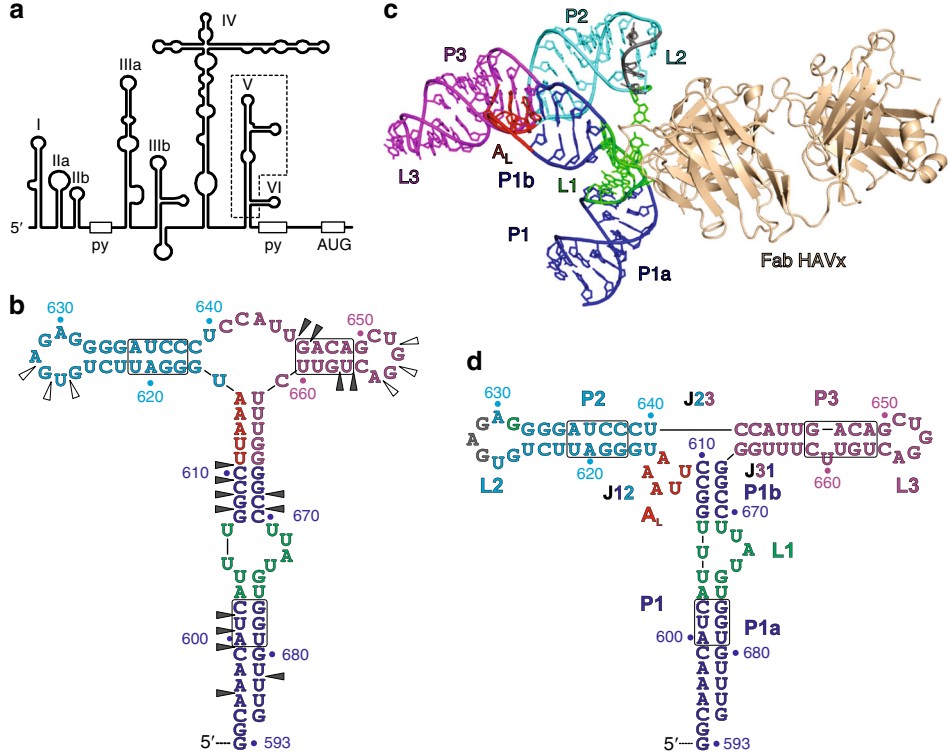

**Fig. 1** Design of the HAV IRES dV crystallization construct and its overall structure in complex with Fab HAVx. **a** Proposed secondary structure of HAV IRES (wild-type HM175 strain)[28] showing major RNA domains (I–VI), polypyrimidine tracts (py) and the start codon (AUG). Dotted box highlights the dV crystallization construct. **b** Proposed secondary structure of the dV crystallization construct according to Brown et al[29]. Empty and filled arrowheads indicate sites of cleavage with dsRNA and ssRNA specific RNases, respectively. Boxed regions indicate the sites of nucleotide covariations. **c** Global structure of the dV–Fab HAVx complex. **d** Secondary structure of dV derived from the crystal structure. Major structural elements and the corresponding nucleotides in **b**–**d** are colored analogously for facile comparison. Nucleotides involved in binding interactions with Fab HAVx and the nucleotides that form an $A_L$ are colored green and red, respectively. Modeling of three nucleotides G627–G629 (colored gray in **c**, **d**) was ambiguous due to poor electron density

## Results

### RNA construct for phage display selection and crystallization.
Domains V and VI reside at the 3′-end of HAV IRES, downstream of the so-called central domain (domain IV) and upstream of the pyrimidine-tract. They contain ~125 nucleotides that form extensive secondary structures (Fig. 1a and Supplementary Figs. 1 and 2). Our RNA construct (Fig. 1b) for phage display selection and crystallization included 92 nucleotides spanning residues 593–684 (nucleotide numbering refers to wild-type HAV strain, HM175)[29] referred to hereafter as HAV domain V or dV. This construct lacks domain VI (nt 681–705 of the WT sequence), which is a small stem-loop inserted near the bottom of the P1 helical stem (as drawn in Fig. 1). In the deletion construct, the final few nucleotides are expected to continue the P1 stem as in the WT structure (Fig. 1, Supplementary Fig. 2). Additionally, for biochemical analysis, we prepared two longer constructs that include both domains V and VI: HAV593-706 and HAV593-720 with the latter construct also containing the pyrimidine tract sequence (Supplementary Figs. 2 and 4).

### Selection of a Fab against dV by phage display.
Previously, we have used phage libraries displaying Fabs derived from the humanized Fab4D5 framework bearing reduced codon diversity in their complementarity determining regions (CDRs) to select Fabs that specifically bind to RNA targets of interest. We subsequently employed these Fabs for RNA crystallization and structure determination[35–37]. To identify Fabs that specifically bind to dV, we performed selections using three reduced codon libraries, termed YSG, YSGR, and YSGRKX. The YSG and YSGR

libraries contained constant sequences in CDRs-L1 and -L2 derived from the parent Fab4D5 sequence, and both libraries contained binary degenerate codon diversity in CDR-L3, -H1, and -H2, encoding equal proportions of Y and S at specific positions. The two libraries differed in their CDR-H3 diversity. In the YSG library, seven residues from the parent Fab4D5 CDR-H3 were replaced with diversified loops of variable lengths (6–17 residues) in which each position was a mixture of 20% Y, 15% S, 15% G, and 50% Z, where Z represents an equimolar mixture of all other natural amino acids except for Y, S, G, and C. In the YSGR library, each CDR-H3 position encodes 38% Y, 25% S, 25% G, and 12% R. In the YSGRKX library all six CDRs were diversified[38]. CDR-L1 and -L2 contained equal proportions of Y and S, CDR-H1 and -H2 contained equal proportions of Y, F, and S, and CDR-L3 and H3 encoded 25% Y, 15% S, 10% G, 12.5% R, 7.5% K and 30% X, where X represents all other amino acids except C, I, and M. Additionally, CDR-L1, -L3, -H1 and -H3 were designed to have variable loop sizes: CDR-L1 (5–6 residues), -L3 (2–8 residues), -H1 (3–8 residues), and -H3 (4–17 residues).

Using each of these libraries, we carried out four rounds of phage display selection against dV RNA. The YSG library produced no Fab binders against this RNA despite having yielded multiple Fabs in previous selections against the P4-P6 domain from the group I intron[35]. Using the YSGR library we obtained several clones that showed a positive RNA binding response in a phage ELISA assay. We expressed these Fabs individually as soluble proteins in *E. coli* using a phagemid expression vector, purified them by affinity and ion-exchange chromatography to obtain RNase free Fabs, and tested their binding affinity for the dV using a filter binding assay. The lowest $K_d$ we observed for the

expressed clones was 290 ± 10 nM (average ± standard deviation, $n \geq 3$). As our previously used Fab crystallization chaperones had RNA binding affinity in the low nanomolar range (10–100 nM), we proceeded with affinity maturation of this Fab using error-prone PCR as described previously[36]. However, we observed no significant improvement in the affinity for the RNA target and did not pursue this Fab further. Using the YSGRKX library we observed that a single clone was 20,000 times enriched after four rounds of selection. Following expression as soluble protein and purification by affinity and ion-exchange chromatography, we found that this Fab bound to dV with good affinity, with a dissociation constant ($K_d$) of 44 ± 8 nM as determined by a filter binding assay in 10 mM tris, pH 7.5, 50 mM KCl, 10 mM $MgCl_2$ buffer at 23 °C (Supplementary Fig. 3). This Fab, designated Fab HAVx hereafter, was advanced to further analysis and crystallization trials. The RNA constructs HAV593-706 and HAV593-720 also bind to the Fab HAVx with similar affinity ($K_d = 49 ± 8$ nM, $47 ± 6$ nM, respectively) compared to the dV construct ($K_d = 44 ± 8$ nM, Supplementary Fig. 4),

**Crystallization and structure determination of the HAVx–dV complex.** To test the ability of Fab HAVx to serve as a chaperone for the RNA crystallization, we set up crystallization trials for the RNA constructs HAV593-684 (or dV), HAV593-706 and HAV593-720 in complex with the Fab HAVx using the hanging drop vapor diffusion method. In 22 out of 480 conditions screened, we observed crystals only for the Fab HAVx–dV complex. We observed no crystals in analogous trials using only the RNA, underscoring the usefulness of the Fab in assisting the dV crystallization. Four conditions were further optimized for pH, precipitant and salt concentration to grow larger crystals using the hanging drop vapor diffusion method. Within 1 week, we observed robust growth of large crystals in 0.2 M ammonium sulfate, 0.1 M HEPES, pH 7.5, 25% PEG 3350 at room temperature that diffracted to 2.84-Å resolution. To solve the crystal structure of Fab–dV complex, we obtained the initial phases by molecular replacement using the previous crystal structure of Fab BL3-6 (PDB code: 4KZE or 3IVK) minus all CDR loops as a search model, highlighting another benefit of Fabs for structure determination. Except for the CDR loops, sequences of the scaffold-domain of the Fab HAVx and Fab BL3-6 (PDB code: 4KZE) are identical; however, due to the flexibility of the Fab elbow angle[39], we searched for the constant domain first followed by the variable domain rather than the intact scaffold. RNA was built unambiguously by modeling individual nucleotides into the electron density map obtained after the initial molecular replacement. After iterative rounds of model building and refinement at 2.84-Å resolution, the final values of $R_{free}$ and $R_{work}$ were 25.2% and 18.6%, respectively. The structural model of the HAVx–dV complex from the final refinement along with the $2|F_o| - |F_c|$ electron density map is shown in Supplementary Fig. 5. Details of data collection and refinement statistics are provided in Table 1.

**Overall structure of the HAVx–dV complex.** The HAVx–dV complex crystallized in the $P2_1 2 2_1$ space group lattice and contained two Fab–RNA complexes per asymmetric unit (Supplementary Fig. 6). Two RNA copies (represented by A and B chains) within the asymmetric unit appear identical except that U659 is unpaired and flipped-out of the helix in chain A, whereas in chain B the flipped-out nucleotide is U660 (Supplementary Fig. 6). The majority of intermolecular interactions in the crystal are mediated by the Fab (Supplementary Fig. 7). Analysis of buried surface area using PDBePISA[40] revealed that including the Fab–RNA binding interface, Fab-mediated contacts account for ~82% of the buried surface area in the crystal packing, suggesting

| **Table 1 X-ray crystallography data collection and structure refinement statistics** | |
|---|---|
| **Data collection** | |
| Space group | $P2_1 2 2_1$ |
| Resolution (Å) | 29.61-2.84 (2.94-2.84) |
| Cell dimensions | |
| a, b, c (Å) | 74.15, 100.58, 236.59 |
| α, β, γ (°) | 90, 90, 90 |
| $R_{merge}$ (%) | 10.0 (128.2) |
| I/σI | 13.90 (1.40) |
| $CC_{1/2}$ | 0.999 (0.60) |
| Completeness (%) | 99.6 (96.5) |
| Redundancy | 6.8 (6.8) |
| **Refinement** | |
| No. reflections | 42, 532 (4057) |
| $R_{work}/R_{free}$ (%) | 18.6/25.2 |
| R.M.S deviations | |
| Bond angles (°) | 1.180 |
| Bond length (Å) | 0.009 |
| Average B-factor, all atoms (Å$^2$) | 100.0 |
| Ramachandran plot of protein residues | |
| Preferred regions (%) | 95.97 |
| Allowed regions (%) | 4.03 |
| Number of residues | |
| RNA | 92 |
| Protein | 876 |
| Solvent | 180 |

Values in the parentheses are for the highest resolution shell

a prominent role of the Fab as a RNA crystallization chaperone. Nevertheless, the observed RNA–RNA contacts, which account for ~18% of the buried surface area in the crystal lattice, also appear to be important in crystal packing. Each RNA molecule in the Fab–RNA complex makes crystal contacts with its three neighboring RNA molecules (Supplementary Fig. 7). Two sets of these crystal contacts involve base-pairing interactions between loop L3 and the 5'-overhanging nucleotides of the neighboring RNA molecule to extend and cap the P1 helical stem. The third set of RNA–RNA crystal contacts occurs between neighboring L2 loops and involves stacking interactions between nucleotides G625 and U626 from symmetry-related molecules. Due to poor electron density for modeling nucleotides G627–G629, other interactions between symmetry-related L2 loops were ambiguous. Interestingly, the chain B RNA (but not the chain A) makes helix-helix contacts with its symmetry-related neighbors. In particular, the 2'-hydroxyl group of A620 makes hydrogen bonding contacts with the 2'-hydroxyl and sugar oxygen of C639, and the 2'-hydroxyl group of U622 interacts with O2 of the flipped-out nucleotide U660.

Overall, dV RNA from HAV IRES folds into a Y-shaped structure (Fig. 1c, d) that consists of two helical stems, P2 and P3 closed by loops L2 and L3, respectively, bifurcated from a base helical stem, P1, formed by base pairing between the 5'- and 3'-ends. These three helical stems, P1, P2, and P3, form a three-way junction where the P2 and P3 helices stack coaxially. The P1 helix protrudes from the P2-P3 helical axis towards the P2 helix with no unpaired nucleotides between P1 and P3. An unpaired region within the P1 helix, L1, forms an asymmetric bulge between the P1a and P1b helical stems. This bulge strongly kinks the RNA backbone, bending helix P1a toward helix P3. (Fig. 1c). The L1 motif constitutes the binding site for Fab HAVx. (Fig. 1c). Nucleotides U611–A615 form a small stem-loop motif ($A_L$) between the P1 and P2 helices, adopting a lone-pair trinucleotide loop (LPTL) structure. The A-rich trinucleotide loop is closed by a single, non-canonical U611•A615 pair (Fig. 1b and d). Overall,

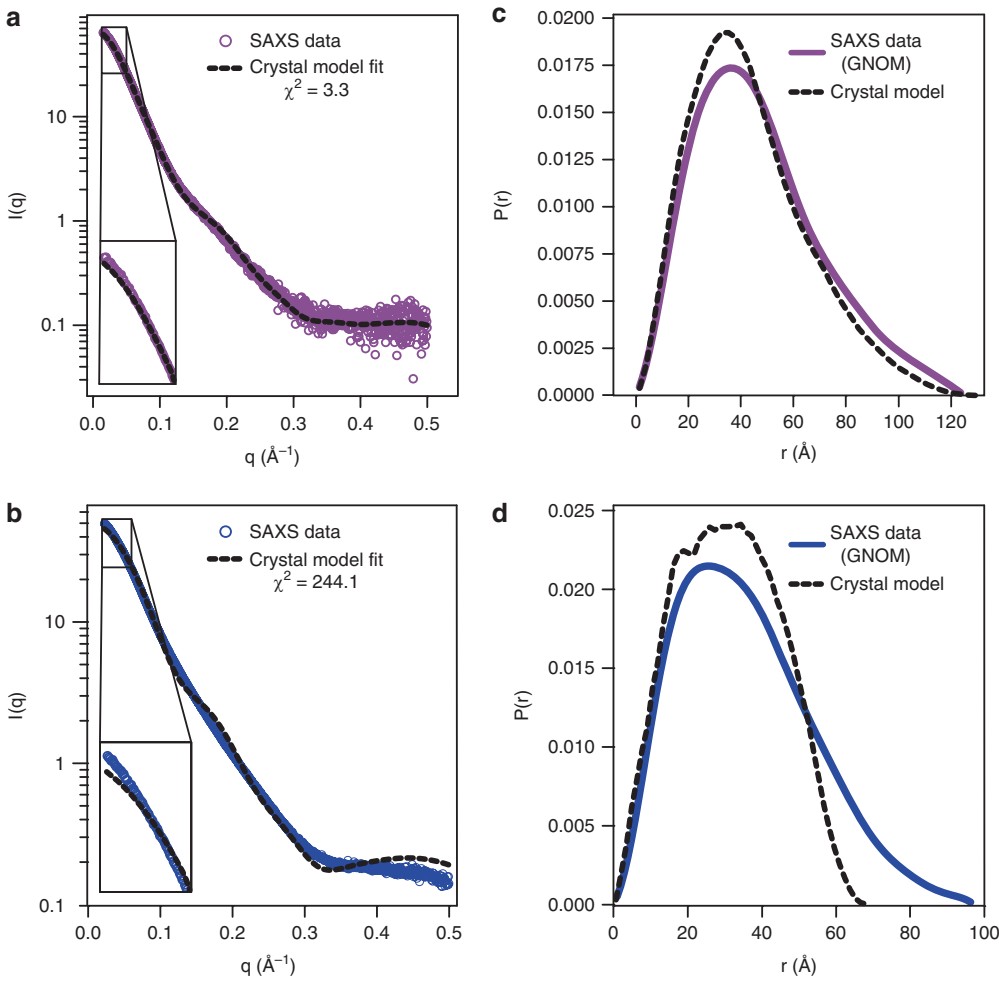

**Fig. 2** Solution phase analysis of the dV–HAVx complex and standalone dV with SAXS. **a** Fitting of experimental SAXS profile (scattering vector, q = 4π sin θ/λ) obtained by merging the datasets from 1 mg/ml, 2 mg/ml and 4 mg/ml concentrations for the dV–HAVx complex and **b** for the standalone dV against the scattering profile calculated from the corresponding crystal structure models. Details of SAXS analysis for both standalone dV and dV–HAVx complex are provided in Supplementary Figs. 7 and 8, and Supplementary Table 1. **c** Pairwise distance distribution calculated from the experimental dataset and the corresponding crystal models for the dV–HAVx complex and **d** for the standalone dV

the secondary structure of the dV from HAV IRES derived from our crystal structure (Fig. 1d) agrees with the previous biochemical data in terms of the paired stems, single-stranded loops and bulged regions (Fig. 1b)[29]. However, our structure differs from the predicted secondary structure in several respects, particularly in the exact location of the three-way junction. Several nucleotides that were predicted to be unpaired (U616, U640–U646 and C661) do engage in base-pairing interactions (Fig. 1b, d), and the existence of the LPTL motif ($A_L$) was not expected—the nucleotides involved (U611–A615) were instead proposed to contribute base-pairing interactions on the 5′-side of the P1 helix (c.f. Fig. 1b, d). We note that because the three-way junction does not interact with the Fab, its position and structure are unlikely to have been influenced by Fab binding.

**Solution analysis of the standalone dV and HAVx–dV complex**. RNA constructs HAV593-706 and HAV593-720, which include domain IV, bind to Fab HAVx with similar affinity compared to standalone dV, indicating that the overall fold of dV including the Fab binding motif remains even in the presence of additional RNA elements downstream of dV. To further test whether our crystal structure corresponds to the solution structure of the standalone RNA and Fab–RNA complex, we performed small-angle X-ray scattering (SAXS) analysis[41]. Details of data

analysis, plots and SAXS-derived parameters are provided in Supplementary Figs. 8 and 9, and Supplementary Table 1). The SAXS scattering profile for the HAVx–dV complex roughly agrees with that calculated from the crystal structure ($\chi^2 = 3.30$, Fig. 2a and Supplementary Fig. 8). Guinier analysis and real-space transformation of the HAVx–dV scattering yield structural parameters (experimental $R_g = 35.6$ Å vs. crystal structure $R_g = 34.6$ Å, and experimental $D_{max} = 125.0$ Å vs. crystal structure $D_{max} = 130.8$ Å) and an overall distance distribution that follow the crystal structure closely (Fig. 2c, Supplementary Fig. 8 and Supplementary Table 1) suggesting that the fold and shape of the Fab–RNA complex in the crystal structure is maintained in solution. The modest discrepancies between the crystal-derived and SAXS-derived particle size parameters for the Fab–RNA complex suggest a slightly less compact structure in solution compared to the crystal, possibly reflecting different extents of helix P1a bending.

The scattering profile for dV RNA alone deviates more seriously ($\chi^2 = 244.1$) from the profile predicted from the crystal structure (Fig. 2b and Supplementary Fig. 8). Structural parameters and the pairwise distance distribution calculated from the data (experimental $R_g = 28.3$ Å vs. crystal structure $R_g = 24.1$ Å, and experimental $D_{max} = 97.2$ Å vs crystal structure $D_{max} = 70$ Å) suggest that the standalone RNA adopts a more extended conformation in solution compared to the crystal

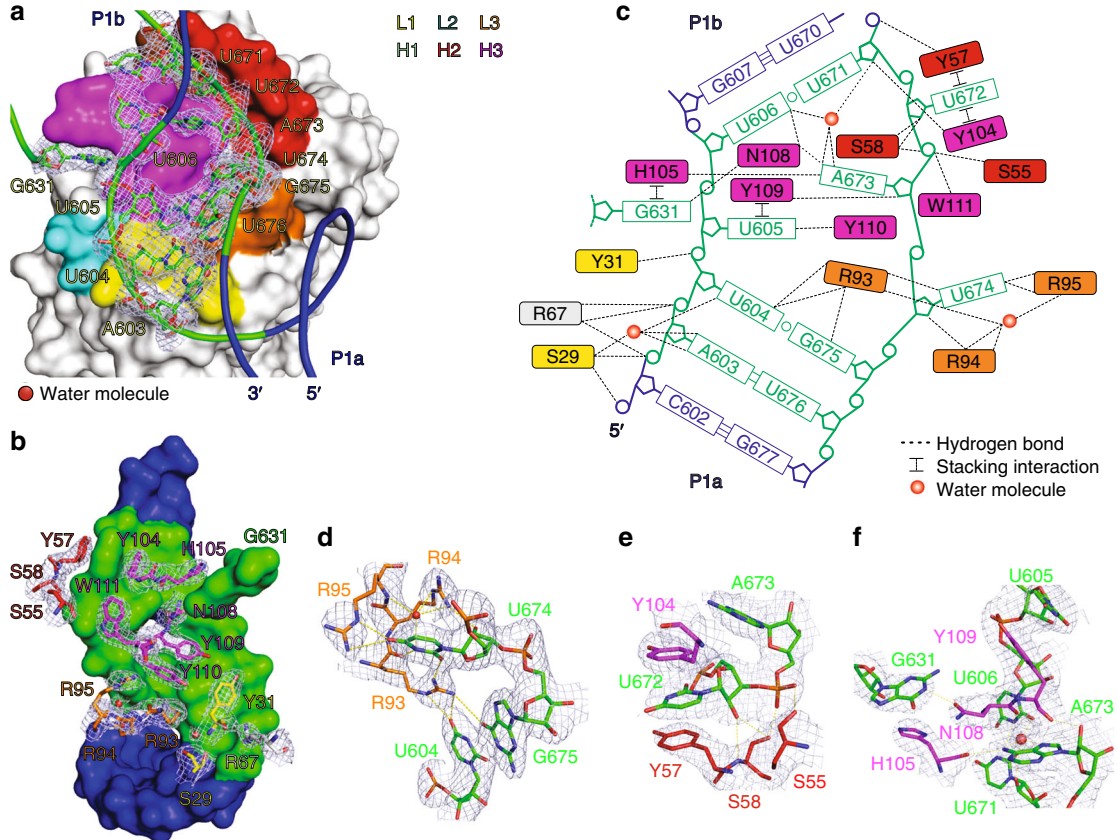

**Fig. 3** Structural features of the dV–Fab binding interface. **a** Molecular surface of the Fab HAVx and a cartoon of the RNA (L1 bulge) showing the Fab CDRs (L1, L2, L3, H1, H2, and H3) and interacting nucleotides of the RNA epitope including G631. **b** Molecular surface of the RNA interface (L1 bulge and G631) and the Fab residues that are involved in binding interactions with the RNA epitope. **c** Schematic summary of Fab–RNA interactions. R67 (gray) represents a scaffold residue from the light chain's constant domain. **d–f** Interactions between the RNA epitope and the Fab CDR residues. Epitope nucleotides and Fab CDR residues are colored analogously in all figures. Orange spheres represent water molecules that mediate hydrogen bonds (dashed lines in **c–f** reflect heteroatoms within hydrogen bonding distance (2.5–3.5 Å). Blue mesh in **a**, **b** and **d–f** represents the $2|F_o|-|F_c|$ electron density map at 1σ contour level and carve radius 1.8 Å

(Fig. 2d, Supplementary Fig. 8 and Supplementary Table 1). In the absence of Fab the unpaired bulge region L1 likely becomes much more dynamic and flexible, potentially allowing helices P1a and P1b to adopt a more coaxial orientation. Crystallization of the RNA–Fab complex but not the RNA alone could reflect, in part, the ability of the Fab to limit the conformational dynamics of the RNA. To investigate this discrepancy further, we calculated a bead model molecular envelope from the dV RNA scattering data (Supplementary Fig. 9). The RNA envelope exhibits the same three-way junction shape as the RNA portion of the Fab–RNA complex structure, but it includes extra volume and is more planar, which is broadly consistent with a more extended conformation of helix P1 and an overall increase in dynamics in the absence of the Fab HAVx. Overall, these data indicate that dV of HAV IRES folds independently and undergoes no major rearrangements in the global structure upon binding to Fab-HAVx. Considering the relatively tight affinity of the Fab-RNA complex, Fab HAVx most likely traps a conformation sampled by dV rather than inducing a high energy structural alteration.

**Structural features of the Fab–RNA binding interface**. Within the binding interface involving Fab CDRs, Fab-RNA interactions bury a total of 1315 Å² of surface area, with the heavy- and light-chain CDRs contributing 819 Å² (62%) and 496 Å² (38%), respectively (PDBePISA)[6]. For comparison, this interfacial area exceeds that observed for Fab-protein complexes (on average

777 ± 135 Å²)[42] and Fab BL3-6–hairpin RNA epitope complex (821 Å²)[43] but approximates that observed for Fab2–P4P6 RNA complex (1316 Å²)[35]. For RNA binding, Fab HAVx predominantly uses arginine, serine and tyrosine residues from four of the six CDRs, L1, L3, H2, and H3 to mediate stacking, electrostatic and hydrogen bond (direct and water-mediated) interactions (Fig. 3a–f). Additionally, R67, a scaffold residue from the light chain's constant domain, makes hydrogen bonding and electrostatic interactions with the RNA phosphate backbone (Fig. 3b, c). While most of the Fab interactions with the RNA localize to the L1 bulge, one nucleotide (G631) from loop L2 also interacts with the Fab (Fig. 3a–c, f). However, this interaction is on the periphery of the interface, and a G631C mutation had only a modest effect on binding to Fab HAVx (Supplementary Fig. 10, $K_d = 58 \pm 12$ nM versus $K_d = 44 \pm 8$ nM for the parent RNA construct), suggesting that this interaction likely has little influence on the stability and conformation of the complex. Supplementary Note 1 contains a detailed description of the interactions within the Fab-RNA interface.

**Structure of the A-rich loop ($A_L$) and the three-way junction**. Our crystal structure of the dV revealed that within the three-way junction, J23 and J31 junction-strands contain no nucleotides whereas the J12 junction-strand contains five nucleotides (U611–A615) (Fig. 4a). This 5-nt UUAAA sequence forms an $A_L$ motif containing a LPTL structure analogous to those observed

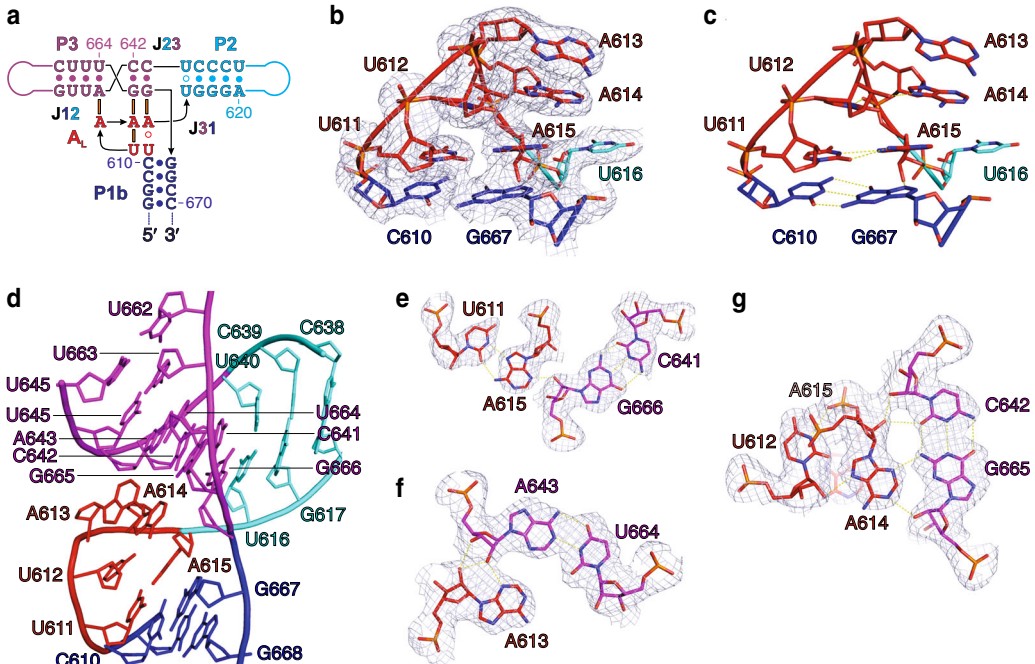

**Fig. 4** Structure of the $A_L$ and organization of the three-way junction. **a** Overall secondary structure of the $A_L$ and the three-way junction. Solid and hollow circles depict canonical and non-canonical base-pairing interactions, respectively, and orange bars indicate the tertiary interactions. **b** Structure of the $A_L$ and **c** interactions within the $A_L$. **d** Structural organization of the three-way junction in which the dinucleotide stack of A613 and A614 from $A_L$ interacts with the minor groove of the helix P3. **e-g** Three base-triples that stabilize the junction involving the $A_L$ nucleotides and the minor groove of the helix P3. Dashed lines in **c** and **e-f** represent heteroatoms within the hydrogen bonding distance (2.5-3.5 Å). Blue mesh in **b** and **e-g** represents the $2|F_o|-|F_c|$ electron density map at 1σ contour level and carve radius 1.8 Å

previously in many rRNA and tRNA structures[32]. The $A_L$ motif (Fig. 4b) stacks directly on the P1 helix and makes extensive A-minor type tertiary interactions with the helix P3, suggesting that the motif plays a significant role in modulating the spatial arrangement of the helices around the three-way junction (Fig. 4a, d–g). In contrast, prior secondary models of dV predicted the $A_L$ sequence (U611–A615) to engage in base-pairing interactions with U662-G666 to extend the P1b helix (Fig. 1b and Supplementary Fig. 2)[29]. The LPTL type of motifs often occurs within the "type C" family of three-way junctions found in rRNA and tRNA, where nucleotides joining non-coaxially stacked helices commonly form "pseudo-hairpin" structures comprising a single base-pair stem and a trinucleotide loop that includes two or three adenines[44]. Typically, the loop nucleotides engage in tertiary interactions with the minor groove of neighboring helices.

Consistent with these LPTL features, the $A_L$ motif in the dV structure forms a single non-canonical U•A pair to close a UAA trinucleotide loop (Fig. 4c) that makes tertiary interactions with the minor groove of P3 (Fig. 4a, d–g). The lone base pair corresponds to a Hoogsteen base pair between U611 and A615, which stacks with the terminal, canonical base pair, C610–G667 of the P1b helical stem (Fig. 4c, e). The trinucleotide loop contains a U-turn between U612 and A613 that creates a sharp directional change in the RNA chain (Fig. 4b–d). The 2′-OH and N3 of U612 form hydrogen bonding interactions with N7 of A614 and the phosphodiester of A615, respectively. Additionally, U612 stacks on U611 whereas A614 forms a stacking sandwich above with A613 and below with A615 of the lone pair (Fig. 4b, c). As observed in different classes of LPTLs[32], a dinucleotide stack, A613 and A614, faces towards the minor groove of the neighboring helix (P3) to mediate A-minor type tertiary interactions (Fig. 4d–g). Overall, three base triples, A613•A643-U664, A614•G665-C642, and A615•G666-C641, involving the LPTL stabilize the three-way junction (Fig. 4e–g).

Strikingly, the architecture and interactions of the $A_L$ motif found in HAV dV structure strongly resemble those of the UUAAA sequence (U1082–A1086) within the GTPase center of E. coli 23S rRNA (PDB code: 1QA6)[45]. In contrast to the Hoogsteen base-pairing in the dV $A_L$ motif, in the ribosomal domain the U and A residues closing the trinucleotide loop form a Watson-Crick base-pair (Supplementary Fig. 12). This $A_L$ motif plays a crucial role in the folding and stabilization of the GTPase center[45], similar to that in HAV dV structure. More broadly, the $A_L$ motif bears an overall resemblance to the structure of the GNRA tetraloop, which mediates interactions with helical minor groove receptors[46,47]. For example, in the GAAA type of GNRA tetraloop observed in the crystal structure of P4–P6 domain of Tetrahymena group I intron (PDB code: 2R8S)[35], the three adenines adopt essentially the same configuration that the three adenines in the $A_L$ motif (A613–A615) adopt, with the A's stacked and oriented analogously for minor groove interactions (Supplementary Fig. 12). In both motifs, the third adenine uses its Hoogsteen face to engage in non-canonical base pairing with upstream residues, forming Sugar Edge/Hoogsteen G•A and Hoogsteen U•A pairs, respectively.

**Similarities between HAV dV and EMCV J–K domain structures.** Despite some differences in the identities of the interacting nucleotides, the global organization of the three-way junction and the $A_L$ mediated base-triple formation revealed in our crystal structure of dV bears a striking resemblance to the high-resolution NMR structure of the J–K domain from EMCV IRES—a type II picornaviral IRES (Fig. 5a–f, details in Supplementary Figs. 12–15)[22]. The J–K domain of EMCV occupies the same relative position within the IRES as does the dV in the HAV IRES. However, the J–K domain resembles a circularly permuted form of dV, where the K, St, and J helices correspond to the P1, P2, and P3 helices of dV and the 5′ and 3′-ends reside at the base

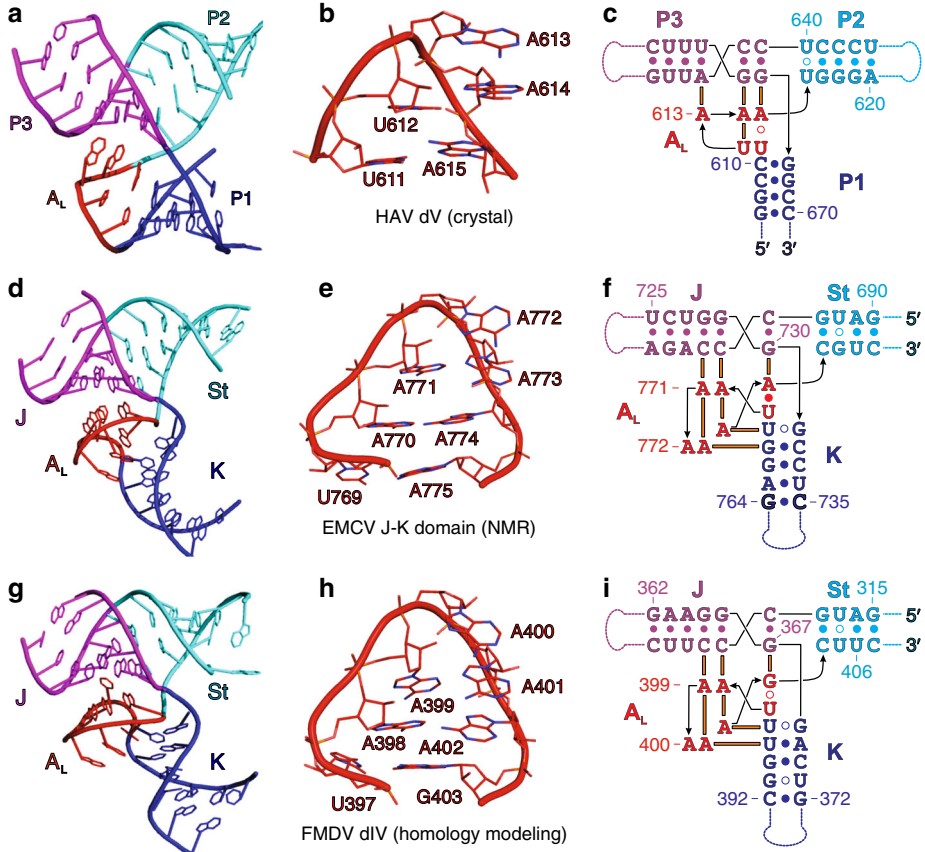

**Fig. 5** Comparison of three-way junction structures of analogous RNA domains within picornaviral IRESs. Three-dimensional structure of the three-way junction, corresponding $A_L$ motif and overall secondary structure of HAV dV (**a–c**), EMCV J–K domain (**d–f**) and FMDV dIV (**g–i**), respectively. Based on the structural homology with EMCV J–K domain, a model of the FMDV dIV structure was computed in silico using the FARFAR and stepwise Monte Carlo Rosetta protocols[55, 56]. For facile comparison, important structural features and the corresponding nucleotides are colored analogously in all figures. Solid and hollow circles depict canonical and non-canonical base-pairing interactions, respectively, and solid orange bars indicate tertiary interactions

of St rather than P1 (Fig. 5a–f, Supplementary Fig. 13). Circularly permuted RNAs can adopt the same overall structure and execute analogous functions. For example, the twister RNA self-cleavage motif and the group II intron occur in circularly permuted forms[48,49]. In the J–K domain, a canonical U769–A775 pair closes the $A_L$ loop, the chain makes a U-turn between A771 and A772, and an A770•A771 dinucleotide stack interacts with the minor groove of the J helix, in a manner analogous to that for the dV (Fig. 5a–f, details in Supplementary Figs. 12–15). However, $A_L$ in the J–K domain contains seven nucleotides, harboring an additional A770:A774 pair between the loop-closing U769:A775 base pair and the A771–A772–A773 triloop such that an additional dinucleotide stack, A772•A773 forms to interact with minor groove of the K helix (details in Supplementary Figs. 12–13). Within the three-way junction, the base triples A770•C695–G729, A771•C696–G728 and A773•G767–C732, A774•U768○G731 connect J and K helical stems, respectively (Supplementary Figs. 14–15). By comparison, we observed the tertiary interactions of $A_L$ with helical stem P3 but not with P2 within the three-way junction of dV structure. Nevertheless, the $A_L$ motif likely locks the coaxial arrangement of P3-P2 and J-St in HAV and EMCV, respectively. Although the J–K domain's $A_L$ does not interact directly with the eIF4G HEAT-1 domain, mutation of all adenines (A770–A775) within the $A_L$ motif to uridines (U770–U775) abrogates eIF4G binding[22]. Similarly, constructs with A771U or C696A–G729U mutations, which prevent A771•C696–G728 base triple formation, do not engage the HEAT-1 domain, implicating the $A_L$ in a functionally significant structural role that determines

the spatial arrangement of the helices around the three-way junction and pre-organizes the J–K domain for recruiting the translation initiation factors[22]. Consistent with the secondary structure homology between the EMCV J–K and FMDV dIV, previous UV-crosslinking and mutation analysis have also shown that deletions or mutations within the corresponding A-rich motif from FMDV IRES abolishes eIF4G binding and reduces IRES activity[50,51].

**Homology modeling of FMDV IRES dIV in silico.** The structural homology associated with HAV dV and EMCV J–K domain led us to examine the sequences of corresponding domains from other picornaviruses for the potential to form $A_L$ motifs[52–54]. For example, previous biochemical assays have shown that EMCV J–K domain and FMDV domain IV (dIV) have homologous secondary structures that bind eIF4G similarly[16,17,33]. Nevertheless, modest rearrangement of the FMDV dIV secondary structure, particularly around the three-way junction, reveals the capacity to form an $A_L$ motif, yielding a revised secondary structure that improves homology between J–K domains, particularly in the J and St stem regions immediately flanking the junction, but remains consistent with the biochemical probing data (c.f. Fig. 5b, c, e, f and Supplementary Fig. 14)[16,17,33]. Based upon this homology we computed a 3-dimensional structure of the FMDV dIV using the FARFAR and stepwise Monte Carlo Rosetta protocols (see Methods)[55,56]. Specifically, we seeded simulations with the three-way junction as well as a two-way junction within the J helix from the EMCV J–K domain, threaded

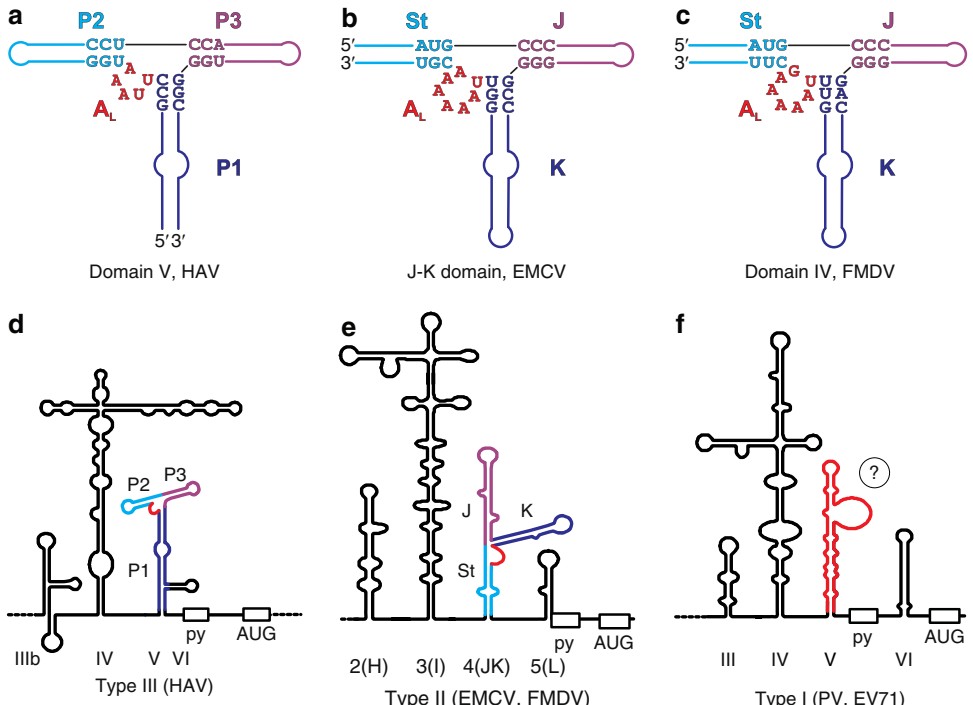

**Fig. 6** Structural homology between the corresponding RNA domains from different types of picornaviral IRESs. **a** Secondary structure of HAV IRES (type III IRES) dV based on the crystal structure obtained in this work. **b** Secondary structure of EMCV IRES (type II IRES) J–K domain based on the previously reported NMR structure (PDB code: 2NBX)[22]. **c** Secondary structure model of FMDV IRES (type II IRES) domain IV computed in silico. The computed model is based upon topological similarities observed between HAV dV and EMCV J–K domain and extensive homology between FMDV dIV and EMCV J–K domain and is consistent with previous biochemical probing results[14, 16, 17, 33]. For facile comparison, important structural features and the corresponding nucleotides have analogous colors. Supplementary information provides detailed comparative structural analysis of HAV dV, EMCV J–K domain, and FMDV dIV (Supplementary Figs. 12–17). **d, e** Secondary structure models of type III and type II picornaviral IRESs, respectively with structurally homologous domains highlighted in red. **f** Available biochemical and bioinformatics data suggests that dV of type I picornaviral IRES also may form a three-way junction with the capacity to form a LPTL type structure (see Supplementary Fig. 18 for details)

with the FMDV sequence and re-optimized. We then solved the remaining junctions and apical loops using stepwise Monte Carlo and used ensembles of those best models to seed a final FARFAR simulation. As expected, the computed model revealed topological homology with EMCV J–K domain and HAV dV, including the similar structural configuration of the three-way junction (Fig. 5g–i and Supplementary Fig. 17). In contrast to the canonical U–A pair in the J–K domain, the closing base-pair in the $A_L$ motif of FMDV dIV consists of a U○G wobble pair. Despite the differences in identity of some nucleotides around the three-way junction, the tertiary interactions of $A_L$ motif with J and K helical stems in both domains are very similar. Briefly, analogous to the base-triple interactions A770•C695–G729, A771•C696–G728, A773•C767–C732 and A774•U768○G731 in EMCV J–K domain, the three-way junction in FMDV dIV is organized and stabilized by A398•C320–G366, A399•C321–G365, A401•U395–A369 and A402•U396○G368 interactions (Fig. 5d–i and Supplementary Figs. 15 and 17). The striking similarity implicates the $A_L$ for overall structural organization of functionally analogous RNA domains within the picornaviral IRESs (Fig. 5c, f and Supplementary Fig. 14).

## Discussion

Comparative analysis between our crystal structure of HAV dV and a previously reported NMR structure of EMCV domain (PDB code: 2NBX)[22] revealed unexpected structural homology in which an $A_L$ motif organizes the topology of the respective three-way junctions (Fig. 5, 6 and Supplementary Figs. 13–15). In addition, both structures contain an asymmetric bulge in

analogous positions (P1 and the K subdomain, respectively; Fig. 6a, b and Supplementary Fig. 13). Based upon the observed structural homology between these domains, we further examined the sequences of corresponding domains from other picornaviruses and revealed that modest rearrangement of biochemically-derived FMDV dIV secondary structure, particularly around the three-way junction has the potential to form an $A_L$ motif[52–54]. Since previous structure probing and biochemical assays have shown that FMDV dIV adopts a secondary structure similar to that of the EMCV J–K domain and that both bind eIF4G similarly[16,17,33], we used the EMCV domain as a structure template together with homology modeling tools to compute a three-dimensional structure model for the rearranged FMDV dIV. The model remained consistent with the biochemical probing data[16,17,33], but improved the structural homology with EMCV J–K domain and HAV dV (Fig. 6a–c), including the tertiary structure of the $A_L$ motif (Fig. 5d–i and Supplementary Figs. 16 and 17). The close similarity of the EMCV and FMDV J–K domains implicates the $A_L$ for overall structural organization of functionally analogous RNA domains within the picornaviral IRESs (Fig. 6a–e). Based on available biochemical, bioinformatic[57] and structural data, domain V of PV IRES (Fig. 6f), a type I picornaviral IRES, also may form a three-way junction possibly with the capacity to form a LPTL type of motif that involves the nucleotides C519–G524 (Supplementary Fig. 18).

The presence of a common motif across picornavirus IRES subtypes implicates a biologically significant function. Consistent with an important contribution from these domains to translation initiation factor recruitment, nucleotide truncations within HAV

dV and EMCV J–K domain decrease or abolish in vitro translation of reporter constructs driven by the corresponding IRES elements. For HAV, the truncations of nts 638–739, 670–739, 638–694, 523–628, and 628–734 have a clear inhibitory effect on translation[23,29,58]. However, the deletion of nts 638–666, which includes the entire P3 helix had almost no effect[23,29,58]. Perhaps the $A_L$ motif still maintains its structure to preserve the overall architecture of the P1 and P2 helices, allowing the recruitment of the translation initiation factors. Although not essential for translation[23,29,58], the highly conserved P3 helical stem possibly has critical roles in other stages of the viral life-cycle. With respect to direct binding to translation initiation factors, Fraser and coworkers[31] recently reported that the IRES (nts 44–737, which includes domains II-VI) binds to eIF4G (aa 682–1599) with $K_d = 239 \pm 10$ nM and eIF4E binding to eIF4G generates an even higher affinity complex that binds with $K_d = 94 \pm 3$ nM. For EMCV, Lomakin et al.[59] reported that the isolated J–K domain binds to eIF4G HEAT-1 domain (aa 643–1076) with $K_d = 5$ nM and 170 nM with and without the eIF4A, respectively. In addition, recent structural studies demonstrated that the HEAT-1 domain binds between the St and K domains[22]. A-rich motif deletion, mutation to a U-rich motif or mutation to perturb the tertiary interactions with the J and K helices abrogate HEAT-1 domain binding to the J–K domain. Each of these mutant constructs retains the secondary structure of the St, J, and K subdomains, implicating a structural role for the $A_L$ motif in modulating the spatial arrangement of the helices around the three-way junction[22]. Consistent with a functionally significant role of the $A_L$ motif, these mutations also have deleterious effects on translation and infectivity of the EMCV[60]. The dV from PV IRES (nts 448–555) binds to eIF4G (aa 557–1599) and eIF4G–eIF4E complex with $K_d$ values of $276 \pm 21$ nM and $49 \pm 2$ nM, respectively[31]. However, analogous data for the isolated HAV dV are not available.

As an alternative to direct binding measurements, translation inhibition assays in the presence of specific RNA constructs can reveal interactions between IRES domains and the translation machinery[61,62]. Addition of isolated EMCV J–K domain or PV IRES dV inhibited in vitro translation of a reporter construct, presumably through sequestration of eIF4G by the isolated viral domain (Supplementary Fig. 19). Addition of HAV324–720 construct that included both dIV and V also inhibited reporter translation but isolated HAV dV had only a little effect (Supplementary Fig. 19). The P1 and P2 helices of the dV might engage eIF4G in a manner analogous to that of the J and St domains; nevertheless, IRES domains and components of the eIF4F complex (eIF4G, eIF4E, and eIF4A) act synergistically and thus, isolated domains from distinct IRES subtypes may not yield the same biochemical signature in the context of a single initiation factor or domain thereof. Moreover, structural and mechanistic differences distinguish the IRES subtypes. For example, in addition to the aforementioned circular permutation of these respective domains, HAV IRES recruits full-length eIF4G and its binding partner eIF4E for translation, whereas other subtypes carry proteinases that cleave eIF4G and utilize for translation the C-terminal product, which lacks the capacity to bind eIF4E[27,63]. Perhaps, the presence of the $A_L$ motif across the picornaviral subtypes reflects conservation of a structural mechanism for organizing the three-way junction in this region of the IRES rather than the preservation of precise molecular details for initiation factor recruitment.

RNA elements within the 3′-UTR of some plant viruses (3′-CITEs) have been implicated in the recruitment of initiation factors for cap-independent translation[9–11]. The 3′-CITE of PEMV2 (pea enation mosaic virus RNA 2) binds eIF4E and eIF4E-eIF4G complex with $K_d = 58 \pm 16$ nM and $48 \pm 21$ nM,

respectively[64]. Analysis of the secondary and computed 3-dimensional structures PEMV2 and PMV (panicum mosaic virus) 3′-CITEs revealed a pre-organized three-way junction that bears some resemblance to EMCV J–K domain or HAV dV except that a pseudoknot rather than an $A_L$ motif dictates the overall junction architecture (Supplementary Fig. 20)[10,34]. This topological similarity, despite high diversity in primary sequences, secondary structures, organizing motifs ($A_L$ or pseudoknot) and the locations within the genome (5′- or 3′-UTR), implicates a structured three-way junction with preorganized helical domains as an effective platform for recruiting translation initiation factors. Viral domain structural information provides valuable input for developing algorithms to predict RNA structures and search for new IRESs and IRES-like RNAs using bioinformatic tools[52,53], testing functional hypotheses, and designing structure-based therapeutics[65–67].

In the crystal structure of the dV–HAVx complex, most of the lattice contacts involved either Fab–Fab or Fab–RNA interactions, underscoring the role of the Fab in facilitating crystallization. Additionally, using one of our previous structures of the Fab scaffold as the molecular replacement model, we could readily obtain the initial phases, thereby circumventing the need for the more tedious and time-consuming traditional phasing approaches such as heavy metal soaking and bromine derivatization of the RNA. As we observed no crystals of the dV RNA in the absence of Fab HAVx, possibly, in addition to facilitating crystal contacts, the Fab limits the dynamic character of the RNA by binding to a specific conformation of the asymmetric bulge. Consistent with this assumption, our SAXS analysis indicates that the standalone RNA adopts a less compact structure in solution compared to the Fab–RNA complex, perhaps reflecting greater flexibility of the L1 bulge in the absence of Fab. In contrast to our previously developed RNA crystallization chaperone, Fab BL3-6, which binds to a terminal stem-loop RNA motif[68], Fab HAVx recognizes an asymmetric bulge within a helix, thereby offering an alternative RNA motif for grafting into an RNA construct to be crystallized with the Fab chaperone. Given the diversity of sequence composition, conformation and length of CDRs, Fabs, in general, could serve as a unique and versatile scaffold with the potential to bind to a wide variety of RNAs (see Supplementary Note 2 for a comparison of Fab HAVx to other RNA binding proteins) for applications beyond chaperone assisted RNA crystallography.

## Methods

**RNA synthesis and purification**. DNA templates for transcription reactions were prepared by PCR amplification of ssDNA oligomer with T7-promotor sequence purchased from integrated DNA technologies (www.idtdna.com). The first two nucleotides of the reverse primer contained 2′-OMe modifications to reduce transcriptional heterogeneity at the 3′ end[69]. RNA was prepared by in-vitro transcription for 3 h at 37 °C in buffer containing 40 mM Tris-HCl pH 7.9, 2 mM spermidine, 10 mM NaCl, 25 mM MgCl$_2$, 10 mM DTT, 30 U/ml RNase Inhibitor, 2.5 U/ml TIPPase, 4 mM of each NTPs, DNA template 30 pmol/ml, 40 µg/ml homemade T7 RNA polymerase[70]. Transcription reactions were quenched by adding 10 U/ml RNase free DNase I (Promega, www.promega.com) and incubating at 37 °C for 30 min. After the Phenol/Chloroform/Isopropanol, pH 4.3 extraction, the RNA was purified by denaturing polyacrylamide gel electrophoresis. The corresponding RNA band was visualized by UV shadowing and excised from the gel. RNA was eluted overnight at 4 °C in 10 mM Tris, pH 8.0, 2 mM EDTA, 300 mM NaCl buffer. The buffer for eluted RNA was exchanged 3 times for pure water using 10 kDa-cutoff size-exclusion column (Amicon, www.emdmillipore.com). RNA was collected, aliquoted and stored at −80 °C until further use.

**Phage display selection**. Phage display to select a Fab that binds the RNA of interest was performed by following similar strategy as described elsewhere[35,36]. For selection, the HAV593-684 construct was used. The RNA construct that contained an additional 3′ overhang sequence, 5′-AGG UCG ACU CUA GAG GAU CCC CGG (x-module) was hybridized with the biotinylated DNA oligonucleotide, 5′-Biotin-ACC GGG GAT CCT CTA GAG TC and this RNA–DNA hybrid were immobilized on the streptavidin-coated magnetic beads via

biotin-streptavidin interaction. For the first round, 500 nM of RNA was immobilized by using a predetermined amount of beads required for complete immobilization and then incubated with $10^{13}$ cfu (colony forming units) of phages for 15 min in 1 ml of selection buffer, PBS (8 mM $Na_2HPO_4$, 1.5 mM $KH_2PO_4$, 137 mM NaCl, and 3 mM KCl), 0.05% Tween 20, 2.5 mM EDTA, 12.5 mM $MgCl_2$, pH 7.4 supplemented with 0.1 mg/ml BSA, 0.1 mg/ml streptavidin, and 1 unit/µL RNase inhibitor (NEB, www.neb.com). The solution was then removed, and the beads were washed twice with the selection buffer. In subsequent rounds, purified phage pools were first incubated with streptavidin beads in the selection buffer for 30 min, and the supernatant was used for the subsequent selection on a King Fisher magnetic particle processor (Thermo Electron Corporation, www.gogenlab.com). The $10^{11}$ cfu of Phages were incubated for 15 min with (50 nM in 2nd and 3rd round and, 5 and 0.5 nM in 4th round) of the RNA in 100 µl of the selection buffer, supplemented with 0.1 mg/ml BSA, 1 unit/µl RNase inhibitor, and 1.5 µM X-module DNA–RNA hybrid as a competitor. Streptavidin magnetic beads were then added to the solution for 15 min to allow the capture of the biotinylated RNA construct together with the bound phages. The beads were then blocked with 50 µM biotin, washed five times with the buffer, and eluted in 50 µL of elution buffer (PBS, and 1 µg/ml biotinylated RNaseA). The biotinylated RNase A was removed from the resulting phage library by incubation with streptavidin beads. After each round of selection, recovered phages were amplified as described previously[35,36]. After 3rd and 4th round of selection, phages were sequenced.

**Fab expression and purification**. Enriched output clones from the 3rd or 4th rounds were tested for binding to target RNA using phage ELISA. For the phage ELISA assay, the RNA construct was immobilized through the x-module via biotin-neutravidin interactions, and ELISA assay was performed according to published protocols[71]. Clones that showed a positive binding response in a phage ELISA assay were reformatted for soluble protein expression with the introduction of a stop codon on phagemids using Q5 site-directed mutagenesis kit (NEB, www.neb.com). Fabs were expressed and purified first on a small scale (100–250 mL culture) as described elsewhere[35,36,71]. No affinity tag was employed in the purification, which is described briefly below. Binding affinity of each clone with the HAV593-684 RNA lacking the x-module was determined by filter binding assay. Fabs that bound the RNA with desirable affinity were then expressed on a larger scale (4-liter culture) according to published protocols[29,30,55]. Both small and large scale Fab production methods essentially followed similar steps and yielded pure and RNase free Fabs. Briefly, phagemids with stop codon were transformed into 55244 chemical competent cells (www.atcc.org) and directly inoculated a starter culture with 100 µg/ml ampicillin. This overnight culture was then used to inoculate 2xYT media and grown for 24 h at 30 °C. Culture was centrifuged at room temperature, cell pellet was resuspended in the same volume of CRAP-Pi media[71] with 100 µg/ml ampicillin and grown for 24 h at 30 °C. Collected cell pellets were lysed in PBS buffer, and Fab proteins were purified using the AKTAxpress fast protein liquid chromatography (FPLC) purification system (Amersham, www.gelifesciences.com) as described previously[35,36]. The lysate in PBS buffer (pH 7.4) was loaded into a protein A column, the eluted Fab in 0.1 M acetic acid was dialyzed back into the buffer PBS (pH 7.4) and loaded into a protein G column. The eluted Fab from protein G column in 0.1 M glycine (pH 2.7) dialyzed to 50 mM NaOAc, 50 mM NaCl buffer (pH 5.5) and loaded into a heparin column. Finally, the eluted Fab in 50 mM NaOAc, 2 M NaCl (pH 5.5) was dialyzed back into 1x PBS (pH 7.4), concentrated, and analyzed by 12% SDS-PAGE using Coomassie Blue R-250 staining for visualization. Aliquots of Fab samples were tested for RNase activity using the RNaseAlert kit (Ambion, www.thermofisher.com). The aliquots of Fab samples were flash frozen in liquid nitrogen and stored at −80 °C until further use.

**Binding affinity measurements**. The binding constants of selected RNA clones and related mutants were determined by nitrocellulose filter binding assay as reported previously[43]. Briefly, ~ 20 pmol of RNA was 5'-$^{32}$P radiolabeled and purified by denaturing polyacrylamide gel electrophoresis. A constant amount of radiolabeled RNA was incubated at 50 °C for 10 min in a buffer containing 10 mM Tris-HCl (pH 7.5), 50 mM KCl, 10.1 mM $MgCl_2$ and 0.1 mM EDTA. The sample was cooled to room temperature for 10 min and incubated for 30 min with Fab HAVx ranging from 2 nM to 2 µM in a final volume of 40 µL. The Bio-Dot apparatus from Bio-Rad was assembled by placing a BA85 nitrocellulose filter (Whatman, www.gelifesciences.com) at the top and Hybond filter at the bottom (Amersham, www.gelifesciences.com) and wells were pre-equilibrated with 100 µL of selection buffer. The Fab–RNA complex was applied and washed 2 times with 100 µl of the selection buffer at a time. Both filters were air dried, exposed to Phosphor-Imager screens, scanned with a Typhoon Trio imager (GE Healthcare) and the amount of RNA retained in each of the filters was quantified by using Image Quant software (Molecular Dynamics). The dissociation constants were calculated by fitting the data of fraction of RNA retained in the nitrocellulose membrane versus the concentration of the Fab to the equation:

$$F = F_0 + F_{max}\left(\frac{[Fab]}{K_d + [Fab]}\right) \quad (1)$$

where $F$ represent the fraction of bound RNA at a given concentration of the Fab, $K_d$ is the dissociation constant and $F_0$ and $F_{max}$ are the minimum and maximum fractions of the bound RNA, respectively.

**Crystallization**. An aliquot of RNA sample was refolded in 10 mM Tris-HCl, pH 7.5, 50 mM KCl, 10.1 mM $MgCl_2$ and 0.1 mM EDTA buffer. For refolding, RNA was heated at 90 °C for 1 min in water, added the appropriate volume of 10× folding buffer and then incubated at 50 °C for 15 min in 1× folding buffer followed by incubation at room temperature for 5 min and in ice for 5 min. The refolded RNA was then incubated with 1.1 equivalents of the Fab at room temperature for 30 min and concentrated to 6 mg/ml using 10 kDa-cutoff, Amicon Ultra-15 column (www.emdmillipore.com). The formation of Fab–RNA complex was confirmed by native polyacrylamide gel electrophoresis (nPAGE). To decrease the number of nucleation events, Fab-RNA complexes were then passed through 0.2 µm cutoff, Millipore centrifugal filter units ((www.emdmillipore.com). A Mosquito liquid handling robot (TTP Labtech, ttplabtech.com) was used to set up high-throughput hanging-drop vapor-diffusion crystallization screens at room temperature using commercially available screening kits from Hampton Research, Sigma and Jena Bioscience. Several trials were reproduced in larger 1 complex µL + 1 µL well hanging drops on siliconized glass slides. Crystals appeared and grew to full size within 2–3 days at 22 °C in a drop with well condition 0.2 M ammonium sulfate, 0.1 M HEPES, pH 7.5, 25% PEG 3350. For cryoprotection, drops containing suitable crystals were brought to 30% glycerol, keeping all other compositions same. Crystals were immediately flash-frozen in liquid nitrogen after being fished out from the drops and taken to Argonne National Laboratory for collecting the X-ray diffraction data.

**Structural data collection, processing, and analysis**. The X-ray diffraction data sets were collected at the Advanced Photon Source NE-CAT section beamlines 244-ID-B and 24-ID-C. All the datasets were then integrated and scaled using its on-site RAPD automated programs (https://rapd.nec.aps.anl.gov/rapd/). Initial phases were obtained by molecular replacement with previously reported structure of Fab BL3-6 (PDB code: 4KZE or 3IVK) as the search model using Phaser on Phenix[72]. Except for the CDRs, sequences of Fab BL3-6 and Fab HAVx are identical. Iterative model building and refinement were performed by using COOT[73], and Phenix package[72]. RNA was built by modeling the individual nucleotides into the electron density map obtained from the molecular replacement. During the refinement, default NCS option in Phenix was selected. Most of the water molecules were automatically determined by Phenix during refinement. Some water molecules were added manually for the positive electron density in the map based on their possibility to form hydrogen bonds with protein or RNA residues. However, we cannot rule out the possibility that at this modest resolution these densities belong to ions like $Mg^{2+}$, $K^+$, $Cl^-$ etc. Solvent-accessible surface area and area of interaction were calculated using PDBePISA (http://www.ebi.ac.uk/pdbe/pisa/). All structure related figures were generated in PyMOL (Schrodinger, www.pymol.org) and figure labels were edited in CorelDraw (Corel Corporation, http://www.corel.com).

**SAXS data collection and analysis**. SAXS experiments were conducted on the SIBYLS beamline at the Advanced Light Source, Lawrence Berkeley National Laboratory following similar protocols described elsewhere[41]. Sample preparation and measurements were performed in the buffer containing 10 mM Tris-HCl, 50 mM KCl, 10.1 mM $MgCl_2$ and 0.1 mM EDTA at pH 7.5. The RNA and Fab–RNA complex samples were prepared and purified and as described above. For each experiment, three different concentration of the samples, 1.0 mg/ml, 2.0 mg/ml and 4.0 mg/ml, in 30 µl were placed in a 96-well plate. SAXS data were collected every 0.3 s with q ranging from 0.015 Å$^{-1}$ to 0.54 Å$^{-1}$ for the total exposure time of 10 s. For blank correction, SAXS data for the buffer were collected both before and after each sample exposure and subtracted from the sample signal. Within each concentration, each buffer-subtracted exposure was checked for radiation damage and any oversaturated points were removed before being averaged together. The final experimental scattering curve was calculated by scaling the averaged data sets for each concentration to the highest concentration (4.0 mg/ml) data set and merging with ALMERGE, extrapolating to infinite dilution. SAXS curves were calculated from the crystal structure atomic coordinates and fit to the experimental data using the FoXS. The bead model molecular envelopes were constructed with DAMMIF. The details regarding the SAXS data collection, scattering-derived parameters, and programs used for the data analysis with associated references are presented in Supplementary Figs. 8, 9 and Supplementary Table 1.

**In silico modeling**. All input files and scripts necessary to conduct the modeling are provided as GitHub repository (https://www.github.com/everyday847/FMDV_homology), along with detailed explanation of each modeling sub-step. All modeling was conducted with Rosetta 3.10[74]. The model from the NMR structure of EMCV (PDB code: 2NBX)[22] was obtained and the central three-way junction was excised, renumbered to match the FMDV numbering (A:692–696 A:728–733 A:766–777 becomes A:317–321 A:365–370 A:394–405), and threaded. The same process yielded a model of the J-arm's central two-way junction (EMCV residues A:698-708 A:716–726 become A:323–333 A:353–363). We conducted two simulations to generate models of the full FMDV Domain IV. First, we directly conducted a FARFAR simulation[55] using the inferred secondary structure previously discussed, seeded with the fixed three-way junction described above; we generated

10,000 models and clustered the lowest-energy 1000 with a cluster radius of 2.0 Å heavy-atom RMSD. Second, we identified several individual junctions and hairpin loops that merited high-resolution modeling: two-way junctions in the St and K helices, as well as the hairpin terminating the J helix. We set up stepwise Monte Carlo simulations[56] for these individual motifs. We generated 10,000 models of each junction, clustered the lowest energy 1000 models with a cluster radius of 2.0 Å heavy-atom RMSD, and took the ten lowest energy cluster centers as representative models of those junctions. Proceeding with those models, we seeded a FARFAR simulation[55] with each of those "libraries" of starting conformations, which generated 40,000 models, and we clustered the lowest energy 1000 models with a cluster radius of 2.0 Å heavy-atom RMSD. We took the most populated cluster center among the ten lowest energy clusters as the single best model of the interaction.

**Reporting summary**. Further information on research design is available in the Nature Research Reporting Summary linked to this article.

## Data availability

The additional data that support the findings of this study are available from the corresponding author upon reasonable request. Atomic coordinates and structure factors for the reported crystal structure have been deposited with the Protein Data Bank under accession number 6MWN. The input files and scripts associated with homology modeling are provided as GitHub repository [https://www.github.com/everyday847/FMDV_homology] as described in methods.

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

## Acknowledgements

This work was supported by grants from the National Institutes of Health (R01AI081987, R01GM102489) and the Chicago Biomedical Consortium with support from the Searle Funds at The Chicago Community Trust to Joseph A. Piccirilli. The crystallographic work is based on research conducted at the Advanced Photon Source on the North-eastern Collaborative Access Team beamlines, 24-ID-B and 24-ID-C, which is supported by a grant from the National Institute of General Medical Sciences (P41 GM103403) from the National Institutes of Health. This research used resources of the Advanced Photon Source, a U.S. Department of Energy (DOE) Office of Science User Facility operated for the DOE Office of Science by Argonne National Laboratory under Contract No. DE-AC02-06CH11357. For SAXS data collection, this work used Advanced Light Source beamline SIBYLS, a US DOE Office of Science User Facility operated for the DOE Office of Science by Lawrence Berkeley National Laboratory under Integrated Diffraction Analysis (IDAT) grant contract DE-AC02-05CH11231. We would like to thank staffs of the Advanced Photon Source at Argonne National Laboratory and Advanced Light Source beamline SIBYLS for providing technical advice during data collection. We are thankful to Dr. Engin Ozkan, University of Chicago for helping with X-ray data processing. We also would like to thank Piccirilli laboratory members, especially to Benjamin Weissman, for critical review of the manuscript.

## Author contributions

D.K. and J.A.P. conceived and designed the experiments. Y.K. performed phage display selection. S.A.S. made essential contribution to Fab expression and purification. D.K. prepared the samples and conducted most of the biochemical, crystallographic and SAXS experiments. DK and Y.S. phased and solved the crystal structure. D.K. analyzed most of the biochemical and crystallization data and interpreted the results with J.A.P and P.A.R. J.F.R. performed the SAXS data analysis. A.M.W. performed in silico modeling and interpreted the results with R.D. D.K, E.V.P, R.D., P.A.R., and J.A.P. analyzed and interpreted the overall data. D.K and J.A.P. wrote the manuscript and edited it based on feedback from P.A.R.

## Additional information

**Competing interests:** The authors declare no competing interests.

**Peer Review Information:** *Nature Communications* thanks Marianna Teplova, Quentin Vicens and Eric Westhof for their contribution to the peer review of this work. Peer reviewer reports are available.

