## [Peer Review File · Nature Communications]

Reviewers' comments:

Reviewer #1 (Remarks to the Author):

Koirala et al. solved the structure of a 92-nt long RNA domain from the HAV genome. This work is important, as very little structural information is currently available for these large IRES elements. A roughly similar structural study, albeit using NMR, was done for a related domain in another virus three years ago (ref. 18).

The authors employed predominantly X-ray crystallography, using an antibody-based method this lab has developed for difficult cases, as well as small-angle X-ray scattering. I am very pleased to see yet another structural feat for this chaperone-assisted crystallization method. This new addition continues to demonstrate the power of this method for tackling difficult cases that may involve somewhat flexible RNAs. The work is overall of high quality (minus some specifics highlighted in the comments below).

Yet, I find that throughout the Results and Discussion of the paper, the focus is more on the role of Fab as a chaperone for crystallization than about the biological relevance of this structure. The sections pertaining to the significance of the structure that the Fab method was able to bring to light only make for a small section of the results and of the discussion. So there is a disconnect here from the way this story was introduced and presented in through the title, abstract and introduction. It is also confusing to me that the domain V of HAV is "related" or "corresponds" (introduction) to the J-K domain in EMCV, yet the authors find it "unexpected" (page 13) that the corresponding three dimensional structure would be similar. If the domains are related we would expect a similar structure, right?

Overall, upon reading this manuscript I find myself hungry for a deeper analysis and I miss the broader perspective. It is certainly very unclear to me why the authors go at length to emphasize the role of Fab as a chaperone for RNA crystallization. I thought that had been sufficiently established through the work this lab has published over the past 10 years, as well as from other work with protein-nucleic acid crystallography, for example of nucleosomes. Starting the discussion with that aspect in particular is to me a bit of a pity. There is untapped potential for having this work shine in a different and more biologically relevant light, and I offer a few suggestions below.

Main general points to address:

- The authors mention that their structure is "T-shaped" and that it was "unexpected". The structure looks more to me like Y-shaped, which is typical of most three-way junctions. P1 is close to P2, as P2 and P3 remain stacked, which is maybe where the T the authors refer to come from? (which is what I could gather from the paragraph on top of page 8). Yet P1 does not come out at a right angle from the P2/P3 stack. It is almost stacked onto P3, but displaced by the AL loop. Furthermore, the overall structure looks like it could be superimposable on models that were generated for a structured element in a plant virus that binds to eIF4E:

<https://www.sciencedirect.com/science/article/pii/S0969212611001377> . Just like the 2Ds of these 3' CITEs are pretty similar to the 2D of the HAV or EMCV domains (for a review see Nicholson & White, Current Opinion Virology 2011, 1:373). So there is actually quite a precedent among virologists to study these structured domains, for unrelated viruses that nonetheless interact with similar translation factors (eIF4E and eIF4G). Hence, I would strongly recommend that the authors analyze their structure in light of this body of work, as they could inform related fields of translation regulation from elements in the 3' end of viral RNAs.

- The authors do an excellent job at describing the Fab-RNA interface, which helps to demonstrate

that the Fab did not induce an unnatural conformation of the three-way junction. But as mentioned above I think the thoroughness of their analysis in that regard also distracts from the key point which should be about the relevance of that structure to foster our understanding of IRES-mediated recruitment of eIF4G, or of translation in general. It seems to me that the Piccirilli lab has already demonstrated that they can use the Fab technique to solve the structures of very different RNAs, which is enough proof in itself that the method works, although I am sure they must have been criticized on that front in the past. Specifically, I am wondering whether the sections on solution analysis and structural features of the Fab-RNA interface should be condensed, moved to SI, etc. I could also see how such a somewhat technical part would gain in visibility by serving as the base for a separate manuscript, perhaps together with similar data for other RNAs that lab has worked with using the Fab method. It would certainly be interesting to start being able to draw patterns from Fab interaction with RNA. But in the current manuscript, especially considering the promise of its title, I'd frankly rather see more of the comparison with EMCV and other IRESes or viral RNA elements (other IRESes?) in the main text figures. The literature is quite abundant for picornaviridae IRESes. One thing that comes to mind is how the structure could for example rationalize prior experiments, carried out perhaps for related IRESes? Along that line, a version of figure S11 would gain in being moved to the main text. Figure 5 just does not cut it, as it shows only 2D structures.

- In continuation of the previous point, I was also wondering whether the authors could use their structure and that of EMCV to model the 3D structures of related domains in other viruses, for example FMDV. They do propose a revised version of the A-rich loop for FMDV, why not propose a model as well? Certainly methods and now programs are out there to predict models with a reasonable reliability (see work by Rhiju Das). On a related note, what could a complex between eIF4G and this domain look like? Here again, the comparison with 3'CITE structures from plant viruses could be worthwhile. A model had also been proposed for example for one such 3' CITE domain and eIF4E. I am thinking something similar could be proposed here, based on footprinting data or other indications of binding sites for eIF4G that would be in the literature.

- Regarding the UUAAA loop, the authors could be more explicit about the fact that the interactions to P3 are of the A-minor type, which have been extensively described. It would help to see whether similar loop-helix interactions occur in the ribosome for example, perhaps in the similar overall context of a three-way junction. Software such as FR3D (Leontis lab) or DrugSite could help to facilitate that search. The conformation of that loop is also reminiscent of that of a typical GAAA adopting a U-turn, so it would be interesting to see a comparison of these two loops. To most people working in the RNA structure field, a figure like figure 4 would be more meaningful if it presented such an analysis. I recall from reference 38 by Lescaute & Westhof that there were instances where A-minor motifs would stabilize the fold of a three-way junction.

Points about structure refinement (thank you for sharing the PDB and map files!):

- The R/Rfree reported in the text and Table 1 do not match.

- The values in the last shell indicate that the resolution is not truly 2.7 angstroms: $1/\sigma$ is way under 2 (it's actually reported to be 0.1 at 2.68 angstroms in the PDB validation report), CC1/2 is very low, and Rmerge is close to 200%; all these are indicators that there are not enough reflections in that bin to provide meaningful information. So I would recommend that the authors refine to maybe 2.8 or 2.9 angstroms. This won't alter their conclusions most likely and it won't affect the quality of their paper, but it will improve the statistics! This may also improve the quite high difference between R and Rfree (6.5%).

- The 6.5% difference between R and Rfree is also likely caused by over-building. I noticed in particular that residues in L2 before G631 have no density at a 1 sigma contour level, and peaks in the

Fo-Fc that indicate the model provides more structural information than the map can support in that particular location (residues 627-629). Geometry of these residues is also poor, as indicated by the PDB validation report, especially for chain B, further highlighting that they can not be properly resolved with the experimental map. B factors are also the highest in that area. I would suggest that the authors try to refine occupancy in that area, or remove the bases but keep the backbone, and see whether that improves the refinement parameters. The authors may also want to use Rcrane (offered within Coot) or Eraser (within Phenix) to correct the geometry of RNA residues while maintaining or improving the fit into density.

- Please show the electron density on figures within the manuscript, such as for the AL loop and for L1.

- Remove "easily" in line 10 on page 7, particularly since the description of the procedure employed that follows may not seem "easy" to a non-specialist...

Minor points:

- box on fig 1a is misleading and should cover only the region shown in the other panels
- too much hype in the last sentence of the introduction: "first high-resolution.." and "first crystal structure..."; although these are true, there are plenty of other RNA elements from viruses whose structures have been solved, and the strategy of dividing a long RNA region into smaller domains that can be studied individually was demonstrated over and over again to lead to stimulating insights. The relevant question for the readers will likely pertain more to what this new structure brings to our understanding of the IRES mechanism, and how that is different between this virus and EMCV or FMDV for example.

- page 9: could the "modest discrepancies" described be due to flexibility in L1 so that it would be bent to a different extent in solution vs crystal, even in the presence of Fab?

Reviewer #2 (Remarks to the Author):

This is an instructive paper on the crystal structure of a piece of a viral IRES. The paper is particularly interesting because it extends the solved structure to other viral IRESes and accordingly refine the secondary structures.

My only problem is with the quality of the molecular structures (the exocyclic groups are not shown in some of them for example). Also, the distances of the contacts are never indicated. How can we assess those then? In Figure 4 (c and f), I have a problem understanding the interactions: in (c) there seems to be contact between O2'(612) and N7(614), but in (f) it is between O2'(612) and N1(614). The authors use also a very approximative nomenclature to annotate those contacts and this does not help the understanding. I see that the resolution is moderate; however, the authors show water molecules (but no distances). For example in Fig. S10, can the authors eliminate the possibility that it is a potassium ion?

Although the paper is already long, it could be discussed that the type of three-way junction illustrated here is the most prevalent one in structured RNAs, esp rRNAs.

Reviewer #3 (Remarks to the Author):

Koirala et al. report co-crystal structure of domain V of hepatitis A virus (HAV) IRES with a synthetic antibody fragment (Fab) that reveals a three-way junction RNA architecture stabilized by a lone pair trinucleotide loop motif. Topologically the structure appears similar to that of the J-K domain of encephalomyocarditis virus (EMCV) IRES reported previously and suggests a conserved biological function of these picornaviral IRES domains in ribosomal initiation complex assembly. Authors used Fab Chaperone-Assisted RNA Crystallography approach to produce diffracting crystals and to solve the structure by molecular replacement using available Fab structure. The authors also analyzed the RNA structures in solution by small-angle X-ray scattering (SAXS) in both the free and chaperone-bound states and concluded that RNA adopts the same overall fold, albeit less compact structure of the free RNA compared to the Fab-RNA complex. Fab targeting domain V binds with high affinity and specifically recognizes the L1 bulge residues of the P1 stem via stacking, electrostatic and hydrogen bonding interactions involving the CDRs side chains. Structural analysis revealed extensive Fab-RNA interface which is likely to induce conformation of the asymmetric bulge in the complex, resulting in bending of P1 stem. The authors also use their structure to revise predicted secondary structures of the corresponding domains of other picornaviruses, emphasizing their potential to form AL motif that determines the relative arrangement of the helices around the three-way junction. Given the lack of high-resolution three-dimensional structures of picornavirus IRES elements, the reported structure of an RNA domain from type III IRES provides an example of a conserved structural motif relevant for IRES activity.

I have some comments for authors to address before publication.

1. Based on the structural similarity with the J-K domain of EMCV IRES which binds the heat domain of eIF4G between the similarly positioned helices, the authors attempted to investigate if dV of HAV IRES engages eIF4G in an analogous manner. In the Discussion they mentioned "rather weak ($>2\mu\text{M}$)" binding estimated for two studied dV constructs and a recombinant eIF4G. However, no experimental details were provided in the Results and Methods sections (i.e. binding assay and protein construct used, measured K_d values with estimated s.d. for each RNA construct). The conclusion is therefore unclear: does domain V directly interact with eIF4G, and if yes, with what binding affinity? Filter binding assay might be inappropriately used for this purpose. Perhaps ITC could be used in this case to reliably measure K_d in the micromolar range and to compare it with the K_d measured for EMCV J-K domain binding to the heat domain of eIF4G ($1.3 \pm 0.1 \mu\text{M}$). It would be also interesting to investigate the impact of mutations or deletions of flipped-out bases in L1 bulge recognized by Fab in the complex structure (U674 and U672) on target protein binding.

2. Crystallographic details should be included such as the number of protein and RNA residues or atoms in the asymmetric unit. RNA model building procedure should be described in Methods, and if NCS included in refinement. Concerning crystallographic data processing, $I/\sigma I$ in the highest resolution shell is below 1, while the generally accepted cut off is 2.0. In addition, $CC(1/2)$ value of 0.34 indicates that high-resolution cut-off might not be appropriately determined. Concerning refinement, the $R_{\text{work}}/R_{\text{free}}$ gap is above 6. This suggests that the structure is suffering from model bias.

3. Experimental methods:

a. RNA synthesis: please provide a reference for the preparation of "homemade T7 polymerase".

b. No description/reference was provided for phage ELISA assay.

c. Fab protein expression and purification: Were any affinity purification tags used? What cell strains were used for expression? Was the same buffer used throughout your purification?

4. Page 11, 1st sentence, "...the three-way junction, J23 and J31 ... J12 junction strand... (Fig. 4a)" needs clarification: J23, J31 and J12 junction strands were not defined and not labeled on Fig. 4a.

5. Page 11, 2d paragraph, last sentence, "...base triples, A613•A643-U664, A614•G665-C642 and A615•G666-C641... (Fig. 3e, f, g)" needs correction: the listed base triples correspond to Fig. 4g, f, e, respectively.

6. Figure panels showing hydrogen bonding interactions (i.e. Fig. 3d, e, f and Fig.4e, f, g, Fig. S10) would benefit from atom type color coding (as in Fig 4c).

7. It would be nice to show 2Fo-Fc omit map for some nucleotides involved in the important intermolecular contacts with Fab or LPTL formation, perhaps as a panel in one supplementary figure.

Response to referees

Reviewer #1 (Remarks to the Author):

Koirala et al. solved the structure of a 92-nt long RNA domain from the HAV genome. This work is important, as very little structural information is currently available for these large IRES elements. A roughly similar structural study, albeit using NMR, was done for a related domain in another virus three years ago (ref. 18).

The authors employed predominantly X-ray crystallography, using an antibody-based method this lab has developed for difficult cases, as well as small-angle X-ray scattering. I am very pleased to see yet another structural feat for this chaperone-assisted crystallization method. This new addition continues to demonstrate the power of this method for tackling difficult cases that may involve somewhat flexible RNAs. The work is overall of high quality (minus some specifics highlighted in the comments below).

Yet, I find that throughout the Results and Discussion of the paper, the focus is more on the role of Fab as a chaperone for crystallization than about the biological relevance of this structure. The sections pertaining to the significance of the structure that the Fab method was able to bring to light only make for a small section of the results and of the discussion. So, there is a disconnect here from the way this story was introduced and presented in through the title, abstract and introduction. It is also confusing to me that the domain V of HAV is “related” or “corresponds” (introduction) to the J-K domain in EMCV, yet the authors find it “unexpected” (page 13) that the corresponding three-dimensional structure would be similar. If the domains are related, we would expect a similar structure, right?

Overall, upon reading this manuscript I find myself hungry for a deeper analysis and I miss the broader perspective. It is certainly very unclear to me why the authors go at length to emphasize the role of Fab as a chaperone for RNA crystallization. I thought that had been sufficiently established through the work this lab has published over the past 10 years, as well as from other work with protein-nucleic acid crystallography, for example of nucleosomes. Starting the discussion with that aspect in particular is to me a bit of a pity. There is untapped potential for having this work shine in a different and more biologically relevant light, and I offer a few suggestions below.

We are thankful to the reviewer for the positive comments and constructive suggestions. In the revised manuscript we have placed less emphasis on the Fab and its role in crystallization

and RNA recognition, and we focused more on dV structure and its implications. Accordingly, we have trimmed down the Fab related results and discussion in the main manuscript and moved the relevant contents to supplementary information. Regarding why we think our findings are unexpected, although previous biochemical data suggested that the dV of HAV may be analogous to the J-K domain of EMCV or to dV of PV because they occupy analogous relative positions in the viral 5'-UTR with respect to the start codon, the proposed secondary structure of HAV dV was quite distinct from EMCV or PV. As such, there was no expectation that their structures would be similar. We have tried to make this clearer in the revised manuscript.

Responses to the specific comments and issues raised are provided in the following.

Main general points to address:

- The authors mention that their structure is “T-shaped” and that it was “unexpected”. The structure looks more to me like Y-shaped, which is typical of most three-way junctions. P1 is close to P2, as P2 and P3 remain stacked, which is maybe where the T the authors refer to come from? (which is what I could gather from the paragraph on top of page 8). Yet P1 does not come out at a right angle from the P2/P3 stack. It is almost stacked onto P3, but displaced by the AL loop.

We agree with the reviewer. The main point was to convey the coaxiality of P2-P3, and as the reviewer points out P1 does not emerge from the P2-P3 co-linear axis at a right angle, we have revised the manuscript in accord with this correct perspective.

Furthermore, the overall structure looks like it could be superimposable on models that were generated for a structured element in a plant virus that binds to eIF4E: <https://www.sciencedirect.com/science/article/pii/S0969212611001377>. Just like the 2Ds of these 3' CITEs are pretty similar to the 2D of the HAV or EMCV domains (for a review see Nicholson & White, Current Opinion Virology 2011, 1:373). So, there is actually quite a precedent among virologists to study these structured domains, for unrelated viruses that nonetheless interact with similar translation factors (eIF4E and eIF4G). Hence, I would strongly recommend that the authors analyze their structure in light of this body of work, as they could inform related fields of translation regulation from elements in the 3' end of viral RNAs.

We thank the reviewer for these insightful suggestions. We have now included a version of the following description in different sections of the manuscript (introduction and discussion) that highlight the comparisons of our dV structure with other RNA domains associated with viral translation including cap-independent translational elements (3'-CITEs) located at the 3'-UTRs of many plant viruses.

The cap-independent translation of many plant viruses involves RNA elements located near or within the 3'-UTRs of their genomes, termed cap-independent translational elements (3'-CITEs).¹⁻³ Despite significant differences in size and location within viral genomes compared to IRES elements, 3'-CITEs essentially play roles analogous to IRES elements in recruiting translation initiation factors or the ribosome subunits.¹⁻³ The 3'-CITEs “circularize” the viral genome presumably by base-pairing interactions with the 5' end, thereby priming the genome for translation initiation.¹⁻³ Considering the structural homology that we observed among the analogous domains of picornaviral IRESs, we sought to compare our structure with 3'-CITEs. Interestingly, the secondary structural models of CITEs from many plant viruses that bind translation initialization factors eIF4E and eIF4G-eIF4E complex suggest that these 3'-CITEs likely form three-way junction structures.^{4,5} Indeed, predicted three-dimensional structure models of the panicum mosaic virus (PMV) and pea enation mosaic virus (PEMV) 3'-CITEs reveal pre-organized, structured three-way junctions that bear some resemblance to EMCV J-K domain or HAV dV.^{4,5} The structure assumes a T-shaped architecture, in which the 5'-3' helix stacks coaxially with one of the other two helices. Residues at the position analogous to the A_L in the EMCV J-K domain or HAV dV form a pseudoknot with the helix positioned at right angle to the co-axially stacked helices.⁴ Like IRES domains, these 3-way junction structures in many plant viruses bind the translation initiation factors or the ribosome subunits with high affinity.¹⁻³ For example, pseudoknot-containing PEMV2 3'-CITE binds to eIF4E and eIF4F complex with $K_d = 58 \pm 16$ nM and $K_d = 48 \pm 21$ nM, respectively.⁴ These comparisons elicit the possibility that despite high diversity in sequences and secondary structures, structured three-way helical junctions within viral RNAs may represent a common topological strategy for highjacking the host translation initiation machinery for cap-independent translation.

- The authors do an excellent job at describing the Fab-RNA interface, which helps to demonstrate that the Fab did not induce an unnatural conformation of the three-way junction. But as mentioned above I think the thoroughness of their analysis in that regard also distracts from the key point which should be about the relevance of that structure to foster our

understanding of IRES-mediated recruitment of eIF4G, or of translation in general. It seems to me that the Piccirilli lab has already demonstrated that they can use the Fab technique to solve the structures of very different RNAs, which is enough proof in itself that the method works, although I am sure they must have been criticized on that front in the past. Specifically, I am wondering whether the sections on solution analysis and structural features of the Fab-RNA interface should be condensed, moved to SI, etc. I could also see how such a somewhat technical part would gain in visibility by serving as the base for a separate manuscript, perhaps together with similar data for other RNAs that lab has worked with using the Fab method. It would certainly be interesting to start being able to draw patterns from Fab interaction with RNA. But in the current manuscript, especially considering the promise of its title, I'd frankly rather see more of the comparison with EMCV and other IRESes or viral RNA elements (other IRESes?) in the main text figures. The literature is quite abundant for picornaviridae IRESes. One thing that comes to mind is how the structure could for example rationalize prior experiments, carried out perhaps for related IRESes? Along that line, a version of figure S11 would gain in being moved to the main text. Figure 5 just does not cut it, as it shows only 2D structures.

We appreciate the reviewer's suggestions. We have trimmed down the contents related to detailed description of the Fab-RNA interactions and moved some of the relevant contents to supplementary information. The revised manuscript focuses on the implications of our structure and its comparison with RNA domains from other IRESs and 3'-CITEs. Specifically, we have revised figures 5 and 6 of the main manuscript for clearer and broader comparison among HAV dV, EMCV J-K domain and FMDV dIV. The revised discussion section "Comparison of dV structure with other viral RNA domains suggests a conserved topology" and related figures in the supplementary information (Fig. S12-S18) further details the comparison.

- In continuation of the previous point, I was also wondering whether the authors could use their structure and that of EMCV to model the 3D structures of related domains in other viruses, for example FMDV. They do propose a revised version of the A-rich loop for FMDV, why not propose a model as well? Certainly, methods and now programs are out there to predict models with a reasonable reliability (see work by Rhiju Das). On a related note, what could a complex between eIF4G and this domain look like? Here again, the comparison with 3'CITE structures from plant viruses could be worthwhile. A model had also been proposed for example for one such 3' CITE domain and eIF4E. I am thinking something similar could be proposed here, based on foot-printing data or other indications of binding sites for eIF4G that would be in the literature.

We thank the reviewer for these constructive suggestions. We collaborated with Rhiju Das and his coworker, Andrew M. Watkins to model FMDV dIV. As expected, the predicted 3D-structure bears strong similarity to the high-resolution NMR structure of EMCV J-K domain. The manuscript now contains a separate results section “Homology modeling of FMDV IRES dIV *in silico*”. Associated methods are included in the “Methods” section. Professor Das and Andrew M. Watkins are coauthors on the revised manuscript. We believe this additional work heightens the impact and broader significance of our findings.

We agree with the reviewer that modeling of the dV and eIF4G complex would be interesting. Based upon the available data, it is not entirely clear that HAV IRES binds to eIF4G in the same manner that the EMCV IRES does, despite the observed topological similarity in analogously positioned domains (also see our response to reviewer 3 for additional information). With the exception of HAV, picornaviruses use encoded proteases to cleave eIF4G, and thereby shut down translation of host mRNAs.⁶ Thus, EMCV IRES can bind the fragments of eIF4G, and reported data indicate that the J-K domain in isolation binds to the HEAT-1 domain of eIF4G (aa 643 – 1076) with $K_d = 170$ nM.⁷ In contrast, translation of the HAV genome requires uncleaved eIF4G.⁸ Results presented below in our response to reviewer 3 (also included in supplementary information Fig. S19) suggest that while the isolated J-K RNA domain from EMCV can inhibit translation of a reporter mRNA, presumably because of its ability to bind to eIF4G and block its translation initiation activity, the isolated dV of HAV cannot achieve such inhibition. With no high-resolution structure of full-length eIF4G available and the lack of clear evidence that dV can bind on its own to eIF4G, this modeling would be conceptually and technically challenging to perform at this stage. However, we have included in the revised manuscript a discussion comparing the structures of these analogous picornaviral IRES domains and 3'-CITEs found in many plant viruses.

- Regarding the UUAAA loop, the authors could be more explicit about the fact that the interactions to P3 are of the A-minor type, which have been extensively described. It would help to see whether similar loop-helix interactions occur in the ribosome for example, perhaps in the similar overall context of a three-way junction. Software such as FR3D (Leontis lab) or DrugSite could help to facilitate that search. The conformation of that loop is also reminiscent of that of a typical GAAA adopting a U-turn, so it would be interesting to see a comparison of these two loops. To most people working in the RNA structure field, a figure like figure 4 would be more

meaningful if it presented such an analysis. I recall from reference 38 by Lescoute & Westhof that there were instances where A-minor motifs would stabilize the fold of a three-way junction.

We thank the reviewer for this suggestion. The revised manuscript now describes the comparison of the dV UUAAA loop (A_L) with other similar motifs including the structure of a GAAA tetraloop. A figure (shown below) related to this comparison is included in supplementary information as Fig. S12.

Figure S12: Comparison of HAV dV A_L motif with other similar motifs and GNRA tetra-loops. Structures of A_L motif in (a) HAV dV (PDB code: 6MWN), (b) EMCV J-K domain (PDB code: 2NBX)⁹ and (c) GTPase associated RNA domain of *E. coli* 23S rRNA (U1082-A1086, PDB code: 1QA6).¹⁰ (d-f) Superposition of A_L motifs from HAV dV and EMCV J-K domain (d), 23S rRNA U1082-A1086 (e), and a GAAA type GNRA tetraloop observed in the crystal structure of P4-P6 domain of *Tetrahymena* group I intron (f, PDB code: 2R8S).¹¹

The following discussion has been added in the main manuscript:

Strikingly, the architecture and interactions of the A_L motif found in our HAV dV structure strongly resemble those of the UUAAA sequence (U1082-A1086) within the GTPase center of

E. coli 23S rRNA (PDB code 1QA6).¹⁰ In contrast to the Hoogsteen base-pairing in the dV A_L motif, in the ribosomal domain the U and A residues closing the trinucleotide loop form a Watson-Crick base-pair (supplementary information Fig. S12). This A_L motif plays a crucial role in the folding and stabilization of the GTPase center,¹⁰ similar to that in HAV dV structure. More broadly, the A_L motif bears an overall resemblance to the structure of the GNRA tetraloop, which mediates interactions with helical minor groove receptors.^{12,13} For example, in the GAAA type of GNRA tetraloop observed in the crystal structure of P4-P6 domain of *Tetrahymena* group I intron (PDB code: 2R8S)¹¹, the three adenines adopt essentially the same configuration that the three adenines in the A_L motif (A613-A615) adopt, with the A's stacked and oriented analogously for minor groove interactions (supplementary information Fig. S12). In both motifs, the third adenine uses its Hoogsteen face to engage in noncanonical base pairing with upstream residues, forming Sugar Edge/Hoogsteen G•A and Hoogsteen U•A pairs, respectively.

Points about structure refinement (thank you for sharing the PDB and map files!):

- The R/Rfree reported in the text and Table 1 do not match.

We apologize for this error. The error has been fixed in the revised manuscript.

- The values in the last shell indicate that the resolution is not truly 2.7 angstroms: I/sigma is way under 2 (it's actually reported to be 0.1 at 2.68 angstroms in the PDB validation report), CC1/2 is very low, and Rmerge is close to 200%; all these are indicators that there are not enough reflections in that bin to provide meaningful information. So, I would recommend that the authors refine to maybe 2.8 or 2.9 angstroms. This won't alter their conclusions most likely and it won't affect the quality of their paper, but it will improve the statistics! This may also improve the quite high difference between R and Rfree (6.5%).

As suggested by the reviewer, we reprocessed the data and refined our structure at 2.84-Å resolution. We updated the Table 1 accordingly. Now, the I/sigma, CC1/2 and Rmerge are 13.9 (1.4), 0.999 (0.60), and 10.0 (128.2), respectively. This reprocessing of the data did not improve the gap between Rwork and Rfree but did improve the other statistics (previous R_{work}/R_{free} (%) = 19.6/26.1 and updated R_{work}/R_{free} (%) = 18.6/25.2).

- The 6.5% difference between R and Rfree is also likely caused by over-building. I noticed in particular that residues in L2 before G631 have no density at a 1 sigma contour level, and peaks

in the Fo-Fc that indicate the model provides more structural information than the map can support in that particular location (residues 627-629). Geometry of these residues is also poor, as indicated by the PDB validation report, especially for chain B, further highlighting that they cannot be properly resolved with the experimental map. B factors are also the highest in that area. I would suggest that the authors try to refine occupancy in that area, or remove the bases but keep the backbone, and see whether that improves the refinement parameters. The authors may also want to use Rcrane (offered within Coot) or Erraser (within Phenix) to correct the geometry of RNA residues while maintaining or improving the fit into density.

As suggested by the reviewer, we also refined the model after removing the nucleotides 627-629, but it did not change the statistics. Therefore, we have kept those nucleotides in the final model and in the manuscript described the uncertainty in modeling those nucleotides. Moreover, we have not drawn any conclusions based on those nucleotides.

- Please show the electron density on figures within the manuscript, such as for the AL loop and for L1.

The revised manuscript now shows the $2F_o - F_c$ map for the nucleotides involved in important interactions within the three-way junction and the Fab-RNA interface. We have also included the $2F_o - F_c$ map for the overall Fab-RNA structure in the supplementary information Figure S5.

- Remove “easily” in line 10 on page 7, particularly since the description of the procedure employed that follows may not seem “easy” to a non-specialist...

This issue has been addressed in the revised manuscript.

Minor points:

- box on fig 1a is misleading and should cover only the region shown in the other panels

We have revised the figure 1a to fix this error.

- too much hype in the last sentence of the introduction: “first high-resolution...” and “first crystal structure...”; although these are true, there are plenty of other RNA elements from viruses whose structures have been solved, and the strategy of dividing a long RNA region into smaller

domains that can be studied individually was demonstrated over and over again to lead to stimulating insights. The relevant question for the readers will likely pertain more to what this new structure brings to our understanding of the IRES mechanism, and how that is different between this virus and EMCV or FMDV for example.

Appreciating the reviewer's comments, we have deleted the text "...of an RNA domain from a type III IRES and the first crystal structure ..." from the introduction of the revised manuscript. The revised manuscript focuses on the implications of our structure and its comparison with RNA domains from other IRESs.

- page 9: could the "modest discrepancies" described be due to flexibility in L1 so that it would be bent to a different extent in solution vs crystal, even in the presence of Fab?

It is possible that binding of the Fab bends the P1a helix differently in solution than in crystal or it could simply be because of the constraints imposed due to the crystal contacts. We have revised the related sentence in the SAXS section of the manuscript as "The modest discrepancies between the crystal-derived and SAXS-derived particle size parameters suggest a slightly less compact structure of the Fab-RNA complex in solution compared to the crystal, possibly reflecting different extents of L1 bending within the Fab-RNA complex in solution versus in the crystal."

Reviewer #2 (Remarks to the Author):

This is an instructive paper on the crystal structure of a piece of a viral IRES. The paper is particularly interesting because it extends the solved structure to other viral IRESes and accordingly refine the secondary structures.

We thank the reviewer for the positive comments and appreciate the time and efforts taken to review the manuscript.

My only problem is with the quality of the molecular structures (the exocyclic groups are not shown in some of them for example). Also, the distances of the contacts are never indicated. How can we assess those then? In Figure 4 (c and f), I have a problem understanding the

interactions: in (c) there seems to be contact between O2'(612) and N7(614), but in (f) it is between O2'(612) and N1(614).

We have revised the structural figures to address these issues. In the revised figures, to clarify the contacts, the atoms are color-coded and the hydrogen bonding distances are indicated in the corresponding figure legends.

The authors use also a very approximative nomenclature to annotate those contacts and this does not help the understanding. I see that the resolution is moderate; however, the authors show water molecules (but no distances). For example, in Fig. S10, can the authors eliminate the possibility that it is a potassium ion?

Most of the water molecules were automatically determined by Phenix during refinement. Some water molecules were added manually for the positive electron density in the map based on their possibility to form hydrogen bonds with protein or RNA residues. However, we cannot rule out the possibility that at this modest resolution these densities belong to ions like Mg^{2+} , K^+ , Cl^- etc. This explanation has been added in the methods under the section “structural data collection, processing and analysis”. Appreciating the reviewer’s suggestion and comments, we have indicated the possible hydrogen bonding distances in the legends of the revised figures.

Although the paper is already long, it could be discussed that the type of three-way junction illustrated here is the most prevalent one in structured RNAs, esp. rRNAs.

Based on this suggestion and comments from the reviewer 1, in the revised manuscript, we have elaborated this discussion. Because we have moved much of the Fab-RNA binding related contents to supplementary information, this does not increase the length of the manuscript.

Reviewer #3 (Remarks to the Author):

Koirala et al. report co-crystal structure of domain V of hepatitis A virus (HAV) IRES with a synthetic antibody fragment (Fab) that reveals a three-way junction RNA architecture stabilized by a lone pair trinucleotide loop motif. Topologically the structure appears similar to that of the J-K domain of encephalomyocarditis virus (EMCV) IRES reported previously and suggests a

conserved biological function of these picornaviral IRES domains in ribosomal initiation complex assembly. Authors used Fab Chaperone-Assisted RNA Crystallography approach to produce diffracting crystals and to solve the structure by molecular replacement using available Fab structure. The authors also analyzed the RNA structures in solution by small-angle X-ray scattering (SAXS) in both the free and chaperone-bound states and concluded that RNA adopts the same overall fold, albeit less compact structure of the free RNA compared to the Fab-RNA complex. Fab targeting domain V binds with high affinity and specifically recognizes the L1 bulge residues of the P1 stem via stacking, electrostatic and hydrogen bonding interactions involving the CDRs side chains. Structural analysis revealed extensive Fab-RNA interface which is likely to induce conformation of the asymmetric bulge in the complex, resulting in bending of P1 stem. The authors also use their structure to revise predicted secondary structures of the corresponding domains of other picornaviruses, emphasizing their potential to form AL motif that determines the relative arrangement of the helices around the three-way junction. Given the lack of high-resolution three-dimensional structures of picornavirus IRES elements, the reported structure of an RNA domain from type III IRES provides an example of a conserved structural motif relevant for IRES activity.

I have some comments for authors to address before publication.

We thank the reviewer for the valuable comments and suggestions. Responses to the specific comments and issues raised are provided in the following.

1. Based on the structural similarity with the J-K domain of EMCV IRES which binds the heat domain of eIF4G between the similarly positioned helices, the authors attempted to investigate if dV of HAV IRES engages eIF4G in an analogous manner. In the Discussion they mentioned “rather weak ($>2\mu\text{M}$)” binding estimated for two studied dV constructs and a recombinant eIF4G. However, no experimental details were provided in the Results and Methods sections (i.e. binding assay and protein construct used, measured K_d values with estimated s.d. for each RNA construct). The conclusion is therefore unclear: does domain V directly interact with eIF4G, and if yes, with what binding affinity? Filter binding assay might be inappropriately used for this purpose. Perhaps ITC could be used in this case to reliably measure K_d in the micromolar range and to compare it with the K_d measured for EMCV J-K domain binding to the heat domain of eIF4G ($1.3 \pm 0.1 \mu\text{M}$). It would be also interesting to investigate the impact of mutations or deletions of flipped-out bases in L1 bulge recognized by Fab in the complex structure (U674 and U672) on target protein binding.

For the viral translation, HAV IRES (type III) requires intact eIF4G in contrast to other picornaviral IRESs (type I and type II) that cleave the eIF4G.^{6,8} For example, 2A^{pro}, 3C^{pro} and L^{pro} proteinases cleaves eIF4G between aa 681 – 682, 712 – 713, 674 – 675 in PV, FMDV and EMCV, respectively. It has been reported that eIF4G (aa 682 – 1599) binds to PV dV with $K_d = 75 \pm 4 \text{ nM}$ ¹⁴ and an eIF4G fragment (aa 643 – 1076) that contains its HEAT-1 domain binds to the isolated EMCV J-K domain with $K_d = 170 \text{ nM}$.⁷ However, there is no report on direct binding of eIF4G with HAV dV in isolation. In an attempt to investigate the binding interactions between eIF4G and HAV IRES dV, we purchased a recombinant eIF4G and performed a filter binding assay. Upon considering the reviewer's comment we realized that the recombinant eIF4G that we purchased contained only aa 1250-1599 of the full-length eIF4G (1599 aa). As HEAT-1 domain containing fragment reported previously that bind to EMCV with $K_d = 1.3 \pm 0.1 \text{ } \mu\text{M}$ consisted of aa 736 – 1115 and the binding assay we described in the original manuscript used purchased eIF4G (1250-1599 aa), which does not support HAV IRES directed translation, the experiment was flawed and the data are not directly comparable. Therefore, we have removed this point from the discussion in the revised manuscript.

Previously published work by Fraser and coworkers¹⁴ reported K_d values for HAV IRES (nts 44 – 737) binding to eIF4G constructs as follows: eIF4G (aa 557-1599), $K_d = 407 \pm 55 \text{ nM}$; (aa 682-1599), $K_d = 239 \pm 10 \text{ nM}$. In the presence of eIF4E, eIF4G (aa 557-1599) binds to the intact HAV IRES (nts 44 – 737) with $K_d = 94 \pm 3 \text{ nM}$. In addition, they demonstrated that the isolated dV of PV IRES (nts 448 – 555) binds to eIF4G (aa 557-1599) and eIF4G (aa 557-1599) – eIF4E complex with K_d values of $276 \pm 21 \text{ nM}$ and $49 \pm 2 \text{ nM}$, respectively. However, analogous data for the isolated dV of HAV IRES have not been reported. We reached out to Dr. Fraser and asked if they had performed these experiments. Dr. Fraser communicated that they did do quite a lot of experiments with the in vitro binding of eIF4G to the PV and HAV IRESes. They tried to narrow down the HAV binding site a little, but noted that they couldn't easily separate the specific binding from non-specific binding using the anisotropy assay, and felt that it would have been a significant undertaking to make various HAV IRES mutants and test them in vitro and in lysate/cells, so they decided not to focus on that in their study. Thus, although HAV dV truncations of nts 638 – 739, 670 – 739, 638 – 694, 523 – 628 and 628 – 734 have a clear effect on translation,¹⁵⁻¹⁷ the role of dV in translation specifically and in the viral lifecycle in general remains unknown.

Appreciating the reviewer's comments, we further pursued experiments designed to examine dV interactions with translation initiation factors (see below Figure R2 or supplementary information Fig. S19). We designed a bicistronic luciferase construct (Figure R2a) for in-vitro translation in which the firefly luciferase expression is controlled by canonical mechanism while renilla expression by hepatitis C virus (HCV) IRES. As HCV IRES binds 40S ribosome directly without requiring any translation initiation factors during translation initiation,¹⁸⁻²⁰ we used this construct to perform rabbit reticulocyte lysate based *in vitro* translation in the presence of various RNA constructs. Added RNA constructs that can bind to the 40S ribosome would be expected to inhibit both firefly and renilla luciferase expression, whereas constructs that bind to eIFs (eIF4G or eIF4E or both) would be expected to inhibit the firefly luciferase expression, and possibly enhance firefly luciferase expression by eliminating competition with the upstream (firefly luciferase) open reading frame for ribosomes. The expression of firefly and renilla luciferase was quantified by measuring the luminescence activity of each protein using a dual reporter assay kit (Promega). As expected, the luminescence signals for both firefly and renilla luciferases decreased to background level by the addition of HCV IRES RNA (2.5 μ M, nts 40 – 372 of genotype 1b^{19,21}) compared to that in the absence of any added RNA (Figure R2b), suggesting that HCV IRES RNA in trans strongly inhibited both firefly and renilla expression, presumably by sequestering 40S ribosomal subunits. Consistently, the addition of IRES domains known to bind eIF4G (PV dV and EMCV J-K domain, 5 μ M),^{9,14} specifically inhibited firefly luciferase translation but not HCV IRES driven translation of renilla luciferase (Figure R2c). In contrast, constructs corresponding to HAV dV alone (nts 593-684, 10 μ M) or HAV dV plus dVI and a portion of polypyrimidine tract (HAV593-720, 10 μ M) had little effect on translation of either luciferase (Figure R2c). However, inclusion of domain IV yielded a construct (HAV324-720, 5 μ M) that exhibited inhibition similar to the EMCV and PV constructs tested (Figure R2c), with an IC50 of 110 ± 30 nM (Figure R2d), consistent with the previous study that full length HAV IRES binds to eIF4G-4E complex with K_d of 94 ± 3 nM.¹⁴

These findings indicate that, in contrast to the ability of EMCV JK domain (or PV dV) to inhibit translation, presumably by sequestering eIF4G or eIF4G-eIF4E complex, the HAV dV domain alone, despite having analogous structural topology to the J-K domain, may not be sufficient to effectively bind the eIF4 complex. This difference in behavior could arise from the different requirements of eIF4G by HAV IRES compared to other picornaviral IRESs. HAV translation requires intact eIF4G but other picornaviruses utilize its cleaved form.^{6,8}

Further to the possible functional role of dV, as noted in our response to reviewer 1, there is a striking similarity between the A_L motif found in our HAV dV structure and a motif formed by UUAAA, (U1082 – A1086) in E. coli 23S rRNA (PDB code: 1QA6)¹⁰ or CUAAG sequence, C1996 – G2000 in human 28S rRNA (PDB code: 4V6X)²² within the GTPase-associated RNA domain of the large ribosomal subunit. In the structure of the human 80S ribosome, this A_L motif engages in an interaction with P0 of the phosphoprotein complex.²² Therefore, we tested whether dV from HAV also has the capacity to bind to ribosomal protein P0. We performed the filter binding assay with recombinant protein (full length P0: 316 aa, purchased from LD Biopharma) in 10 mM tris, pH 7.5, 50 mM KCl, 10 mM MgCl₂ buffer at 23°C, showing that dV binds the P0 with 220 ± 18 nM affinity. Although we are pursuing experiments to delineate whether this newly discovered interaction has functional significance in the HAV lifecycle, we prefer not to publish this observation until these experiments have been completed. In addition, given the challenges that the field has experienced associated with study of the HAV IRES relative to other picornavirus IRES elements, these studies could require significant investment of time and effort.

The revised manuscript includes the major points from above discussion in the section “Comparison of dV structure with other viral RNA domains suggests a conserved topology” and Figure R2 shown below has been now included as supplementary information Fig. S19.

Figure R1: Secondary structure models of the domains of 5'-UTR of the wild-type HAV strain, HM-175.¹⁵ The dV crystallization construct is highlighted red.

Figure R2. *In vitro* translation of a bicistronic luciferase construct in rabbit reticulocyte lysate. (a) Design of the DNA template for the translation. The construct was generated via PCR of pFR_HCV_xb plasmid, which was a gift from Phillip A. Sharp (Addgene plasmid # 11510, <http://www.addgene.org/11510/>),²³ and the translation assay was performed by using a coupled transcription-translation kit (www.promega.com). In the assay, the firefly luciferase expression is controlled by canonical mechanism while renilla expression by hepatitis C virus (HCV) IRES. As HCV IRES binds 40S ribosome directly without requiring any translation initiation factors during translation initiation,¹⁸⁻²⁰ the added RNA constructs that can bind to the 40S ribosome would be expected to inhibit both firefly and renilla luciferase expression, whereas constructs that bind to

eIFs (eIF4G or eIF4E or both) would be expected to inhibit the firefly luciferase expression, and possibly enhance firefly luciferase expression by eliminating competition with the upstream (firefly luciferase) open reading frame for ribosomes. The expression levels of the luciferases were detected by measuring the luminescence signals (Synergy Neo2 plate reader, www.biotek.com), which were obtained by using a dual-luciferase reporter assay (www.promega.com) (b) Normalized luminescence corresponding to firefly and renilla luciferase expression in the presence of HCV IRES RNA (2.5 μ M, nts 40-372 of genotype 1b^{19,21}) compared to that in the absence of any added RNA. (c) Luminescence corresponding to firefly luciferase expression normalized against the renilla luminescence in the presence of analogous RNA domains from HAV, PV and EMCV IRESs. Addition of IRES domains known to bind eIF4G (PV dV and EMCV J-K domain, 5 μ M),^{9,14} specifically inhibited firefly luciferase translation but not HCV IRES driven translation of renilla luciferase. In contrast, constructs corresponding to HAV dV alone (nts 593-684, 10 μ M) or HAV dV plus dVI and a portion of polypyrimidine tract (HAV593-720, 10 μ M) had little effect on translation of either luciferase. However, inclusion of domain IV yielded a construct (HAV324-720, 5 μ M) that exhibited inhibition similar to the EMCV and PV constructs tested. (d) Suppression of firefly luminescence corresponding to the dose dependent inhibition of firefly luciferase expression by HAV 324 -720 construct with an IC₅₀ of 110 ± 30 nM, consistent with the previous study that full length HAV IRES binds to eIF4G-4E complex with K_d of 94 ± 3 nM.¹⁴ The error bars represent the standard deviations from three independent experiments.

2. Crystallographic details should be included such as the number of protein and RNA residues or atoms in the in the asymmetric unit. RNA model building procedure should be described in Methods, and if NCS included in refinement. Concerning crystallographic data processing, I/σ in the highest resolution shell is below 1, while the generally accepted cut off is 2.0. In addition, CC (1/2) value of 0.34 indicates that high-resolution cut-off might not be appropriately determined. Concerning refinement, the R_{work}/R_{free} gap is above 6. This suggests that the structure is suffering from model bias.

As suggested by this reviewer and also by the reviewer 1, we reprocessed the data and refined our structure at 2.84-Å resolution. Although reprocessing of the data improved other statistics, it did not improve the gap between R_{work} and R_{free} (previous R_{work}/R_{free} (%) = 19.6/26.1 and updated R_{work}/R_{free} (%) = 18.6/25.2). The Table 1 has been updated to reflect the changes. It includes number of atoms present in the macromolecular chains. The RNA model building

procedure has been clarified in the manuscript. During the refinement, the NCS was selected automatically in Phenix which has been now indicated in the methods.

3. Experimental methods:

a. RNA synthesis: please provide a reference for the preparation of “homemade T7 polymerase”.

A reference, “Rio, D.C. Expression and purification of active recombinant T7 RNA polymerase from *E. coli.*, *Cold Spring Harbor Protocols*, pdb. prot078527 (2013)” regarding the preparation of T7 RNA polymerase has been added in the revised manuscript.

b. No description/reference was provided for phage ELISA assay.

The immobilization of the RNA construct via biotin-neutravidin interactions and the phage ELISA assay was performed by following the protocols described in “Paduch, M. et al. Generating conformation-specific synthetic antibodies to trap proteins in selected functional states, *Methods*, 60, 3-14 (2013)”. This reference has been added in the revised manuscript.

c. Fab protein expression and purification: Were any affinity purification tags used? What cell strains were used for expression? Was the same buffer used throughout your purification?

We have revised the methods section of the manuscript to make these issues clear including the following specific points.

No purification tags were used. We used Protein-A and protein-G based affinity chromatography columns. These proteins are known to bind constant scaffold region of a Fab. We used *E. coli* 55244 cell strain (www.atcc.org/products/all/55244.aspx). Buffers were varied during purification depending on the column used but after the final round of purification, Fab was stored in PBS buffer. The revised manuscript describes the details of the Fab expression and purification protocol in the methods section. In addition, a citation, “Paduch, M. et al. Generating conformation-specific synthetic antibodies to trap proteins in selected functional states, *Methods*, 60, 3-14 (2013)”, related to this method has been added in the revised manuscript.

4. Page 11, 1st sentence, "...the three-way junction, J23 and J31 ... J12 junction strand... (Fig. 4a)" needs clarification: J23, J31 and J12 junction strands were not defined and not labeled on Fig. 4a.

In the revised manuscript, we have labelled the junction strands in the Figure 1d and 4a.

5. Page 11, 2d paragraph, last sentence, "...base triples, A613•A643-U664, A614•G665-C642 and A615•G666-C641... (Fig. 3e, f, g)" needs correction: the listed base triples correspond to Fig. 4g, f, e, respectively.

We apologize for this error and thank the reviewer for catching this. We have cited the correct figures in the revised manuscript.

6. Figure panels showing hydrogen bonding interactions (i.e. Fig. 3d, e, f and Fig.4e, f, g, Fig. S10) would benefit from atom type color coding (as in Fig 4c).

We have revised the figures and atoms are color coded to clarify the important interactions. We have also indicated the distances for the hydrogen bonding contacts in the legends of corresponding figures.

7. It would be nice to show $2F_o - F_c$ omit map for some nucleotides involved in the important intermolecular contacts with Fab or LPTL formation, perhaps as a panel in one supplementary figure.

We appreciate the reviewer's suggestion. The revised manuscript figures show the $2F_o - F_c$ map for the nucleotides involved in important interactions within the three-way junction and the Fab-RNA interface. The $2F_o - F_c$ map for the overall Fab-RNA complex is shown in supplementary information Figure S5.

Response Letter References

1. Nicholson, B.L. & White, K.A. 3' Cap-independent translation enhancers of positive-strand RNA plant viruses. *Curr. Opin. Virol.* **1**, 373-380 (2011).
2. Simon, A.E. & Miller, W.A. 3' Cap-Independent Translation Enhancers of Plant Viruses. *Ann. Rev. Microbiol.* **67**, 21-42 (2013).
3. Nicholson, B.L., Zaslaver, O., Mayberry, L.K., Browning, K.S. & White, K.A. Tombusvirus Y-shaped translational enhancer forms a complex with eIF4F and can be functionally replaced by heterologous translational enhancers. *J. Virol.* **87**, 1872-1883 (2013).
4. Wang, Z., Treder, K. & Miller, W.A. Structure of a viral cap-independent translation element that functions via high affinity binding to the eIF4E subunit of eIF4F. *Journal of Biological Chemistry* **284**, 14189-14202 (2009).
5. Wang, Z., Parisien, M., Scheets, K. & Miller, W.A. The cap-binding translation initiation factor, eIF4E, binds a pseudoknot in a viral cap-independent translation element. *Structure* **19**, 868-880 (2011).
6. Lloyd, R.E. Translational control by viral proteinases. *Virus research* **119**, 76-88 (2006).
7. Lomakin, I.B., Hellen, C.U. & Pestova, T.V. Physical association of eukaryotic initiation factor 4G (eIF4G) with eIF4A strongly enhances binding of eIF4G to the internal ribosomal entry site of encephalomyocarditis virus and is required for internal initiation of translation. *Molecular and Cellular Biology* **20**, 6019-6029 (2000).
8. Borman, A.M. & Kean, K.M. Intact eukaryotic initiation factor 4G is required for hepatitis A virus internal initiation of translation. *Virology* **237**, 129-136 (1997).
9. Imai, S., Kumar, P., Hellen, C.U., D'Souza, V.M. & Wagner, G. An accurately pre-organized IRES RNA structure enables eIF4G capture for initiating viral translation. *Nat. Struct. Mol. Biol.* **23**, 859 (2016).
10. Conn, G.L., Draper, D.E., Lattman, E.E. & Gittis, A.G. Crystal Structure of a Conserved Ribosomal Protein-RNA Complex. *Science* **284**, 1171-1174 (1999).
11. Ye, J.-D. et al. Synthetic antibodies for specific recognition and crystallization of structured RNA. *Proc. Natl. Acad. Sci. U.S.A.* **105**, 82-87 (2008).

12. Correll, C.C. & Swinger, K. Common and distinctive features of GNRA tetraloops based on a GUAA tetraloop structure at 1.4 Å resolution. *Rna* **9**, 355-363 (2003).
13. Fiore, J.L. & Nesbitt, D.J. An RNA folding motif: GNRA tetraloop–receptor interactions. *Quarterly reviews of biophysics* **46**, 223-264 (2013).
14. Avanzino, B.C., Fuchs, G. & Fraser, C.S. Cellular cap-binding protein, eIF4E, promotes picornavirus genome restructuring and translation. *Proceedings of the National Academy of Sciences* **114**, 9611-9616 (2017).
15. Brown, E.A., Day, S.P., Jansen, R.W. & Lemon, S.M. The 5' nontranslated region of hepatitis A virus RNA: secondary structure and elements required for translation in vitro. *Journal of virology* **65**, 5828-5838 (1991).
16. Brown, E.A., Zajac, A.J. & Lemon, S.M. In vitro characterization of an internal ribosomal entry site (IRES) present within the 5' nontranslated region of hepatitis A virus RNA: comparison with the IRES of encephalomyocarditis virus. *Journal of virology* **68**, 1066-1074 (1994).
17. Glass, M.J., Jia, X.-Y. & Summers, D.F. Identification of the hepatitis A virus internal ribosome entry site: in vivo and in vitro analysis of bicistronic RNAs containing the HAV 5' noncoding region. *Virology* **193**, 842-852 (1993).
18. Fraser, C.S. & Doudna, J.A. Structural and mechanistic insights into hepatitis C viral translation initiation. *Nature Reviews Microbiology* **5**, 29-38 (2007).
19. KIEFT, J.S., ZHOU, K., JUBIN, R. & DOUDNA, J.A. Mechanism of ribosome recruitment by hepatitis C IRES RNA. *Rna* **7**, 194-206 (2001).
20. Otto, G.A. & Puglisi, J.D. The pathway of HCV IRES-mediated translation initiation. *Cell* **119**, 369-380 (2004).
21. Brown, E.A., Zhang, H., Ping, L.-H. & Lemon, S.M. Secondary structure of the 5' nontranslated regions of hepatitis C virus and pestivirus genomic RNAs. *Nucleic acids research* **20**, 5041-5045 (1992).
22. Anger, A.M. et al. Structures of the human and Drosophila 80S ribosome. *Nature* **497**, 80 (2013).
23. Petersen, C.P., Bordeleau, M.-E., Pelletier, J. & Sharp, P.A. Short RNAs repress translation after initiation in mammalian cells. *Molecular cell* **21**, 533-542 (2006).

REVIEWERS' COMMENTS:

Reviewer #1 (Remarks to the Author):

I wish to commend the authors for scrupulously addressing the points raised in my first review of their manuscript. Thank you for your attitude that demonstrates the worth of the peer reviewing system in improving manuscripts.

A question that came up while reading this thoroughly revised version is the existing biochemical evidence precisely for the presence of the A-rich motif. I feel that the current discussion with that regard is somewhat hidden or diluted (lines 382, 439, 457, fig S16, S18). As indeed there is a lot of probing data available for this and similar IRES elements (as mentioned in the introduction, line 64), and probably mutagenesis as well, then it would make sense to show more explicitly how well the revised model for how the HAV IRES is structured fits with that data. For example, would the authors argue that the A-rich domain could be maintained in the 638-666 deletion mutant? Could the authors illustrate how well available probing data match the revised model? This would further corroborate the importance of the A-rich region, without having for example to carry out further mutagenesis or luciferase assays (that would remove the bulge, force its pairing as a helix, etc) in order to show that its presence is real and biologically significant. I would add such a small paragraph somewhere around pages 10-12, or in the discussion.

A few minor comments:

- Table 1: X-ray statistics have improved, but the 6.6% difference between R and Rfree – typically resulting from “over-building”- could most likely be reduced by removing a significant portion of the 180 solvent atoms, that are unlikely to be justified at a 2.84 angstrom resolution. It's OK to leave unassigned blobs of density, as they may correspond to water molecules, or sodium ions, etc., without the possibility to distinguish among these (as correctly mentioned at line 630).
- page 14, line 428: to me, that's where the discussion should start. The key point revealed by the structure is the presence of that A-rich loop.
- figure S12: very interesting! what is the topology around the ribosomal loop inside the GTPase center though? is it also part of a 3-way junction?
- Fig 5: I would add labels for “HAV”, “EMCV”, “FMDV” directly on the figure, for ease of understanding without having to refer to the legend; you may also want to specify that only the top one is a crystal structure, and that the others are computationally-generated models...
- how about an SI figure to show the “3'CITE ...structure that bears some resemblance to EMCV J-K”? (new paragraph starting at line 485)

Reviewer #2 (Remarks to the Author):

I thank the authors for their extended answers. I feel the MS is now improved by this thorough revision.

Reviewer #3 (Remarks to the Author):

The revised manuscript by Koirala et al. is improved significantly by further experiments designed to examine dV interactions with translation initiation factors and by the reprocessing the data and further refinement of the structure at 2.84-Å resolution. The discovery that dV from HAV also has the capacity

to bind to ribosomal protein P0 is intriguing and lays the groundwork for future functional and mechanistic studies. Authors address my concerns in full and I recommend publication.

Point-by-point response to the reviewers

Reviewer #1 (Remarks to the Author):

I wish to commend the authors for scrupulously addressing the points raised in my first review of their manuscript. Thank you for your attitude that demonstrates the worth of the peer reviewing system in improving manuscripts.

A question that came up while reading this thoroughly revised version is the existing biochemical evidence precisely for the presence of the A-rich motif. I feel that the current discussion with that regard is somewhat hidden or diluted (lines 382, 439, 457, fig S16, S18). As indeed there is a lot of probing data available for this and similar IRES elements (as mentioned in the introduction, line 64), and probably mutagenesis as well, then it would make sense to show more explicitly how well the revised model for how the HAV IRES is structured fits with that data. For example, would the authors argue that the A-rich domain could be maintained in the 638-666 deletion mutant? Could the authors illustrate how well available probing data match the revised model? This would further corroborate the importance of the A-rich region, without having for example to carry out further mutagenesis or luciferase assays (that would remove the bulge, force its pairing as a helix, etc.) in order to show that its presence is real and biologically significant. I would add such a small paragraph somewhere around pages 10-12, or in the discussion.

We are thankful to the reviewer for the positive comments and constructive suggestions and for the time and effort taken to review the manuscript.

In the previous version of the manuscript, the discussion of the existing biochemical data associated with HAV IRES dV and EMCV IRES J-K domain was spread through different parts of the manuscript.

For HAV dV, we have discussed a comparison of the biochemically-derived and crystal-derived secondary structures in the last paragraph under the title “overall structure of the HAVx – dV complex” as follows.

Overall, the secondary structure of the dV from HAV IRES derived from our crystal structure (Fig. 1d) agrees with the previous biochemical data in terms of the paired stems, single-stranded loops and bulged regions (Fig. 1b).¹ However, our structure differs from the predicted secondary structure in several respects, particularly in the exact location of the three-way junction. Several nucleotides that were predicted to be unpaired (U616, U640 – U646 and C661) do engage in base-pairing interactions (Fig. 1b and d), and the existence of the LPTL motif (A_L) was not expected – the nucleotides involved (U611 – A615) were instead proposed to contribute base-pairing interactions on the 5'-side of the P1 helix (c.f. Fig. 1b and d).

For EMCV J-K domain, the last paragraph under the title “similarities between HAV dV and EMCV J-K domain structures” discussed the following points:

Nevertheless, the A_L motif likely locks the coaxial arrangement of P3-P2 and J-St in HAV and EMCV, respectively. Although the J-K domain’s A_L does not interact directly with the eIF4G HEAT-1 domain, mutation of nucleotides within the A_L abrogates the eIF4G binding, implicating an indirect structural role of A_L that determines the spatial arrangement of the helices around the three-way junction.²

Appreciating the reviewer’s suggestion, we have now included more explicit discussion of available biochemical probing data relevant to the A-rich motif in corresponding domains of HAV, EMCV and FMDV IRESs. Following are the changes in the revised manuscript.

The last paragraph under the title “similarities between HAV dV and EMCV J-K domain structures” has been modified as follows:

Although the J-K domain’s A_L does not interact directly with the eIF4G HEAT-1 domain, mutation of all adenines (A770 – A775) within the A_L motif to uridines (U770 – U775) abrogates eIF4G binding.² Similarly, constructs with A771U or C696A-G729U mutations, which prevent A771•C696-G728 base triple formation, do not engage the HEAT-1 domain, implicating the A_L in a functionally significant structural role that determines the spatial arrangement of the helices around the three-way junction and pre-organizes the J-K domain for recruiting the translation initiation factors.² Consistent with the secondary structure homology between the EMCV J-K and FMDV dIV, previous UV-crosslinking and mutation analysis have also shown that deletions or mutations within the corresponding A-rich motif from FMDV IRES abolishes eIF4G binding and reduces IRES activity.^{3,4}

In the revised manuscript, the following text has been added in the discussion section.

However, the deletion of nts 638 – 666, which includes the entire P3 helix, had almost no effect.^{1,5,6} Perhaps the A_L motif still maintains its structure to preserve the overall architecture of the P1 and P2 helices, allowing the recruitment of the translation initiation factors. Although not essential for translation,^{1,5,6} the highly conserved P3 helical stem possibly has critical roles in other stages of the viral life-cycle.

For EMCV, Lomakin et al.⁷ reported that the isolated J-K domain binds to eIF4G HEAT-1 domain (aa 643 – 1076) with K_d = 5 nM and 170 nM with and without the eIF4A, respectively. In addition, recent structural studies demonstrated that the HEAT-1 domain binds between the St and K domains.² A-rich motif deletion, mutation to a U-rich motif or mutation to perturb the tertiary interactions with the J and K helices abrogate HEAT-1 domain binding to the J-K domain. Each of these mutant constructs retains the secondary structure of the St, J and K

subdomains, implicating a structural role for the A_L motif in modulating the spatial arrangement of the helices around the three-way junction.² Consistent with a functionally significant role of the A_L motif, these mutations also have deleterious effects on translation and infectivity of the EMCV virus.⁸

A few minor comments:

- Table 1: X-ray statistics have improved, but the 6.6% difference between R and R_{free} – typically resulting from “over-building”- could most likely be reduced by removing a significant portion of the 180 solvent atoms, that are unlikely to be justified at a 2.84 angstrom resolution. It’s OK to leave unassigned blobs of density, as they may correspond to water molecules, or sodium ions, etc., without the possibility to distinguish among these (as correctly mentioned at line 630).

Thanks to the reviewer for the suggestion. Appreciating the reviewer’s comment, we refined the structural models by allowing Phenix to automatically assign the water molecules. With all other refinement parameters being identical, this assignment reduced the number of water molecules to 50. However, the R_{work} and R_{free} were 19.0% and 25.6%, respectively. The difference (6.6%) could be due to some ambiguity in modeling of some flexible part of both Fab and RNA with relatively less defined electron density. As this new refinement did not improve the difference between R_{work} and R_{free} compared to that reported in the previous version of the manuscript, at this stage, we prefer not to change the statistics.

- page 14, line 428: to me, that’s where the discussion should start. The key point revealed by the structure is the presence of that A-rich loop.

Thanks to the reviewer for the suggestion. The revised version of the discussion starts with the comparison of the dV structure with other viral RNA domains.

- figure S12: very interesting! what is the topology around the ribosomal loop inside the GTPase center though? is it also part of a 3-way junction?

The ribosomal A-rich loop is also a part of the three-way junction.^{9,10} However, the topological arrangement of the helices around the loop is different compared to that of HAV IRES dV. Unlike in the dV structure, in the ribosomal three-way junction, extensive tertiary interactions occur between two of the three helices.^{9,10} As mentioned in previous response to the reviewer 3, the A-rich motif in the structure of the human 80S ribosome engages in an interaction with P0 of the phosphoprotein complex.¹⁰ Surprisingly, HAV dV bound to ribosomal protein P0 with ~200 nM affinity. We are pursuing experiments to delineate whether this newly discovered interaction has functional significance in the HAV lifecycle, which will be reported in due course.

- Fig 5: I would add labels for “HAV”, “EMCV”, “FMDV” directly on the figure, for ease of understanding without having to refer to the legend; you may also want to specify that only the top one is a crystal structure, and that the others are computationally-generated models...

Thanks to the reviewer for the suggestion. The revised Figure 5 includes those labels.

- how about an SI figure to show the “3’CITE ...structure that bears some resemblance to EMCV J-K”? (new paragraph starting at line 485).

Appreciating the reviewer’s suggestion, we have now included a Supplementary Figure 20 (also shown below) comparing the structures of PEMV2 3’-CITE with EMCV IRES J-K domain and HAV IRES dV.

Supplementary Figure 20: Comparison of secondary structures of HAV IRES dV (a), EMCV IRES J-K domain (b) and PEMV2 3’-CITE (c). The secondary structures of the dV (crystal) and J-K domain (NMR)² were derived from the respective high-resolution structures, whereas PEMV2 3’-CITE represents a computationally calculated model.¹¹ Despite differences in sequence and strategy to organize the RNA structure, the overall 3-way junction topology of the EMCV J-K and PEMV2 3’-CITE is similar.

Reviewer #2 (Remarks to the Author):

I thank the authors for their extended answers. I feel the MS is now improved by this thorough revision.

We thank the reviewer for the positive comments and constructive suggestions, and for the time and effort taken to review the manuscript.

Reviewer #3 (Remarks to the Author):

The revised manuscript by Koirala et al. is improved significantly by further experiments designed to examine dV interactions with translation initiation factors and by the reprocessing the data and further refinement of the structure at 2.84-Å resolution. The discovery that dV from HAV also has the capacity to bind to ribosomal protein P0 is intriguing and lays the groundwork for future functional and mechanistic studies. Authors address my concerns in full and I recommend publication.

We thank the reviewer for the positive comments and constructive suggestions, and for the time and effort taken to review the manuscript.

References

1. Brown, E.A., Day, S.P., Jansen, R.W. & Lemon, S.M. The 5' nontranslated region of hepatitis A virus RNA: secondary structure and elements required for translation in vitro. *Journal of virology* **65**, 5828-5838 (1991).
2. Imai, S., Kumar, P., Hellen, C.U., D'Souza, V.M. & Wagner, G. An accurately pre-organized IRES RNA structure enables eIF4G capture for initiating viral translation. *Nat. Struct. Mol. Biol.* **23**, 859 (2016).
3. Saleh, L. et al. Functional interaction of translation initiation factor eIF4G with the foot-and-mouth disease virus internal ribosome entry site. *Journal of general virology* **82**, 757-763 (2001).
4. De Quinto, S.L. & Martinez-Salas, E. Interaction of the eIF4G initiation factor with the aphthovirus IRES is essential for internal translation initiation in vivo. *Rna* **6**, 1380-1392 (2000).

5. Brown, E.A., Zajac, A.J. & Lemon, S.M. In vitro characterization of an internal ribosomal entry site (IRES) present within the 5' nontranslated region of hepatitis A virus RNA: comparison with the IRES of encephalomyocarditis virus. *Journal of virology* **68**, 1066-1074 (1994).
6. Glass, M.J., Jia, X.-Y. & Summers, D.F. Identification of the hepatitis A virus internal ribosome entry site: in vivo and in vitro analysis of bicistronic RNAs containing the HAV 5' noncoding region. *Virology* **193**, 842-852 (1993).
7. Lomakin, I.B., Hellen, C.U. & Pestova, T.V. Physical association of eukaryotic initiation factor 4G (eIF4G) with eIF4A strongly enhances binding of eIF4G to the internal ribosomal entry site of encephalomyocarditis virus and is required for internal initiation of translation. *Molecular and Cellular Biology* **20**, 6019-6029 (2000).
8. Hoffman, M.A. & Palmenberg, A.C. Mutational analysis of the J-K stem-loop region of the encephalomyocarditis virus IRES. *Journal of virology* **69**, 4399-4406 (1995).
9. Conn, G.L., Draper, D.E., Lattman, E.E. & Gittis, A.G. Crystal Structure of a Conserved Ribosomal Protein-RNA Complex. *Science* **284**, 1171-1174 (1999).
10. Anger, A.M. et al. Structures of the human and Drosophila 80S ribosome. *Nature* **497**, 80 (2013).
11. Wang, Z., Parisien, M., Scheets, K. & Miller, W.A. The cap-binding translation initiation factor, eIF4E, binds a pseudoknot in a viral cap-independent translation element. *Structure* **19**, 868-880 (2011).